

# The possible 5.9 years oscillation identification from superconducting gravimeter observations

Wei Luan[1,2], Hao Ding[1]

[1]School of Geodesy and Geomatics, Hubei LuoJia Laboratory, Wuhan University, 430079, Wuhan, China
[2]Wuhan Gravitation and Solid Earth Tides National Observation and Research Station, 430071, Wuhan, China

*Correspondence to*: Hao Ding (dhaosgg@sgg.whu.edu.cn)

Keywords: 5.9 years oscillation, SG observations, spectra analysis, Love numbers, core origin

**Abstract.** Surface gravity changes aroused by the periodic ∼5.9 years oscillation (referred to as SYO) are important for understanding its origin and may further constrain Earth's deep interior dynamics, but such signals have not been directly
observed. In this study, we combine multiple spectra methods to analyze six usable superconducting gravimeter (SG) residual series in Western Europe, Canada, and Australia between 1996 and 2019; and try to extract the possible SYO signals from surface gravity observations. The amplitudes of the recovered possible SYO gravity changes vary from 0.5 to 0.9 µGal at different observatories. Comparisons with a derived time-varying gravity model indicate that the phases of gravity SYO may also have a spherical harmonic $Y_{22}$ spatial distribution. The corresponding amplitude transform factors $\delta/h$ between the
observed and modeled signals for different SG stations are about 2.9, greater than the ratio of ∼1.9 for the corresponding tidal Love numbers. The observed amplitudes are also quite different from the predictions of the possible mechanisms suggested by previous studies. Although the SYO is believed to have originated from core motions, our findings mean that the potential physical mechanism should be much more complicated than any existing one. We suggest that the MAC waves arising from the interplay between Magnetic, Archimedes, and Coriolis forces could be a possible excitation source of the SYO. We believe
our gravity observation results should help interpret the SYO in the future.

## 1 Introduction

The ∼5.9 years oscillation was first found in the length of day variation (ΔLOD) and thought to closely correlate with the secular variations in the Earth's core (Liao and Greiner-Mai, 1999; Abarca del Rio et al., 2000; Mound and Buffett, 2003,
2006; Gillet et al., 2010; Holme and de Viron, 2005, 2013; Ding and Chao, 2018a). This intradecadal fluctuation in different observations has been intensely studied of its possible geophysical excitation and complex frequency for over 20 years (e.g., Mound and Buffett, 2003, 2006; Gillet et al., 2010; Holme and de Viron, 2005, 2013; Chao et al., 2014; Ding and Chao, 2018a; Ding, 2019; Chao and Yu, 2020; Ding et al., 2021). Seeing about the above literature, the SYO's period has been estimated to



different values within ~5.8-6 years, which show good agreement with each other in the estimated error ranges; moreover, the
average SYO amplitude in ΔLOD in the 1962-2019 timespan was determined to be ~0.12 ms.

Some dynamic processes in the Earth's core were frequently suggested as the origin of the SYO signal in ΔLOD. Mound and
Buffett (2003, 2006) indicated that the SYO most likely arises from the mantle-inner core gravitational (MICG) coupling.
Chao (2017) further equated the MICG axial torsional libration to the steady SYO in ΔLOD and calculated the corresponding
MIGG strength (torsion constant), which lies in the range given by Davies et al. (2014) from a broad range of viscous mantle
flow models with density anomalies inferred from seismic tomography. Gillet et al. (2010) explained the SYO signal by the
fast torsional waves throughout the fluid outer core based on the core angular momentum (CAM) variation, while Asari and
Wardinski (2012) suggested that the CAM variation is not yet sufficient for arguing that Earth's 6 years CAM oscillation is
robustly resolved by magnetic observation. Gillet et al. (2015) showed that the ΔLOD at 4 to 9.5 years periods could be
explained by torsional waves, which may be triggered by the nonlinear interaction between the magnetic field and sub-decadal
nonzonal motions within the fluid outer core. In addition, Holme and de Viron (2005, 2013) suggested that the SYO in ΔLOD
may be related to or excited by the so-called geomagnetic "jerks", defined as sudden changes in the second-order derivative
of the field. Silva et al. (2012) identified a 6 years periodic signal in the secular acceleration of two geomagnetic field models
and showed that this signal seems to be closely related to some geomagnetic jerks. However, the conjecture of the geophysical
mechanism of the SYO excited by the geomagnetic jerks has been repeatedly checked and doubted in some works (e.g., Cox
et al., 2016; Ding, 2019; Ding et al., 2021). Up to now, the physics mechanism of the SYO is still inconclusive.

In recent years, the fluctuation characteristics and excitations of the SYO have also been investigated using some continuous
and long-span geophysical/geodetic observations, including the polar motion (PM, Ding et al., 2019, 2021; Chen et al., 2019),
GPS (Global Positioning System) displacements (Ding and Chao, 2018a; Watkins et al., 2018; Rosat et al., 2021), geomagnetic
fields (Ding and Chao, 2018a), and gravity-field satellite laser ranging (SRL, Chao and Yu, 2020; Rosat et al., 2021). By
applying the OSE (Optimal Sequence Estimation) array processing technique to the global GPS displacement and geomagnetic
data, Ding and Chao (2018a) revealed that the SYO signal manifests as a westward rotary propagating wave of the sectoral
spherical-harmonic pattern of degree-2 order-2 ($Y_{22}$), which may stem from the sectoral $Y_{22}$ density anomalies or the equatorial
ellipticities in the inner core and the lower mantle, by recognizing the MICG coupling mechanism for the inner-core axial
libration. On this basis, Ding et al. (2020) further constructed a time-varying 3-D displacement model of the SYO and
concluded that the SYO signal gives rise to a maximum surface vertical displacement of up to 1.69 mm. Chao and Yu (2020)
analyzed the $\Delta C_{22}$ and $\Delta S_{22}$ (degree-2 order-2 Stokes coefficients of the time-variable Earth's gravitational field) data series
derived from the SRL measurements over 1992-2018. Their results showed that the ~5.9 years variations in the degree-2 order-
2 Stokes coefficients of gravity are $\sim 2 \times 10^{-11}$, corresponding to the upper-mantle vertical displacement of ~1.1 mm. Although
Rosat et al. (2021) found that the surface deformations for the ~5.9 years frequency band did not exceed 0.8 mm based on 83
JPL (Jet Propulsion Laboratory) residual GPS vertical time series, their results also showed that the maximum amplitude of



the ~5.9 years signal is larger than ~1.6 mm based on 63 IGS (International GNSS Service) residual GPS vertical time series (see their Fig. 3b). Besides, Rosat et al. (2021)'s reanalysis on the SRL measurements indicated that interannual variations in
the degree-2 Stokes coefficients of the gravity field do not exceed $2 \times 10^{-11}$. Note that the Stokes coefficients determined from the SRL observations only capture the global scale redistribution of mass and its associated deformation of the Earth (Rosat et al., 2021; Chen et al., 2022).

In this study, we will try to identify and extract the possible SYO gravity signal from the surface gravity changes observed by
the SGs worldwide, and to confirm whether it also satisfies a $Y_{22}$ spatial pattern (including amplitude and phase).

## 2 Data and processing

The Global Geodynamics Project administered by the International Geodynamics and Earth Tides Service (IGETS) recently provided 62 available observation sequences from 43 SG stations (some of the SGs with a dual sensor, see Supplement Fig.
S1). Here we chose the records with lengths around or longer than 18 years (about three cycles of the SYO period) and rejected relatively poor-quality records with too many or too long gaps. Finally, we selected six appropriate SG records from the CA (Cantley, Canada), CB (Canberra, Australia), MB (Membach, Belgium), MC (Medicina, Italy), MO (Moxa, Germany; record mo034-1 was used), and ST (Strasbourg, France) stations. In this study, the Code "h2" database (processed by staff at the International Center for Earth Tides) at hourly sampling intervals is adopted for the used records, whose detailed information
is summarized in Supplement Table S1. Each time series contains a number of steps and gaps mainly due to instrument maintenance, power failure, and actual geophysical causes (such as earthquakes and strong rainfall). These interferences could result in incorrect identification of random-walk processes, which are well-known to perturb the determinations of the amplitudes and phases of long-period Earth deformation signals (Wang and Herring, 2019; Santamaria-Gomez and Ray, 2021). We will remove some well-known signals before further modifying those steps and gaps.


The SG data pretreatment procedure is illustrated in Fig. 1 by taking the CB record as an example. We first removed the data spikes, and the solid and ocean tides were removed using the corresponding local synthetic tide models from the ETERNA harmonic analysis. Furthermore, the atmospheric and non-tidal oceanic loading effects were subtracted according to the surface gravity variation models from the Ecole et Observatoire des Sciences de la Terre (EOST) Loading Service (courtesy of J.P.
Boy). Next, the most important thing for the pretreatment is the modification of the offsets, which determines whether our final results are correct. The pole motion and rotation rate variation (ΔLOD) effects, which would be removed late, will be used as references for preliminary modifying the offsets. After removing the data spikes, local synthetic tide, atmospheric and non-tidal oceanic loading effects, we first piecewise smoothed the SG residuals according to the visible offsets (marked by the vertical green lines in Fig. 1b) and then changed the step values to best fit to the PM and ΔLOD time series. Further SG
residuals are obtained by removing the pole tides and ΔLOD effects (Fig. 1c). Then, to detect and remove some relatively





smaller offsets (with amplitudes larger than ~4 nm/s$^2$), we combined a low-pass filter (0.8 cpy as the cutoff frequency) and a 1-D median filter (300 days as the window) as shown in Fig. 1d (see Supplement Fig. S2 for detailed description). After that, we bridged the gaps in the residuals by applying linear or spline interpolation and removed the linear trend (Fig. 1e). We finally decimated the SG residuals to the daily data, and all selected SG residual time series are exhibited in Fig. 2.


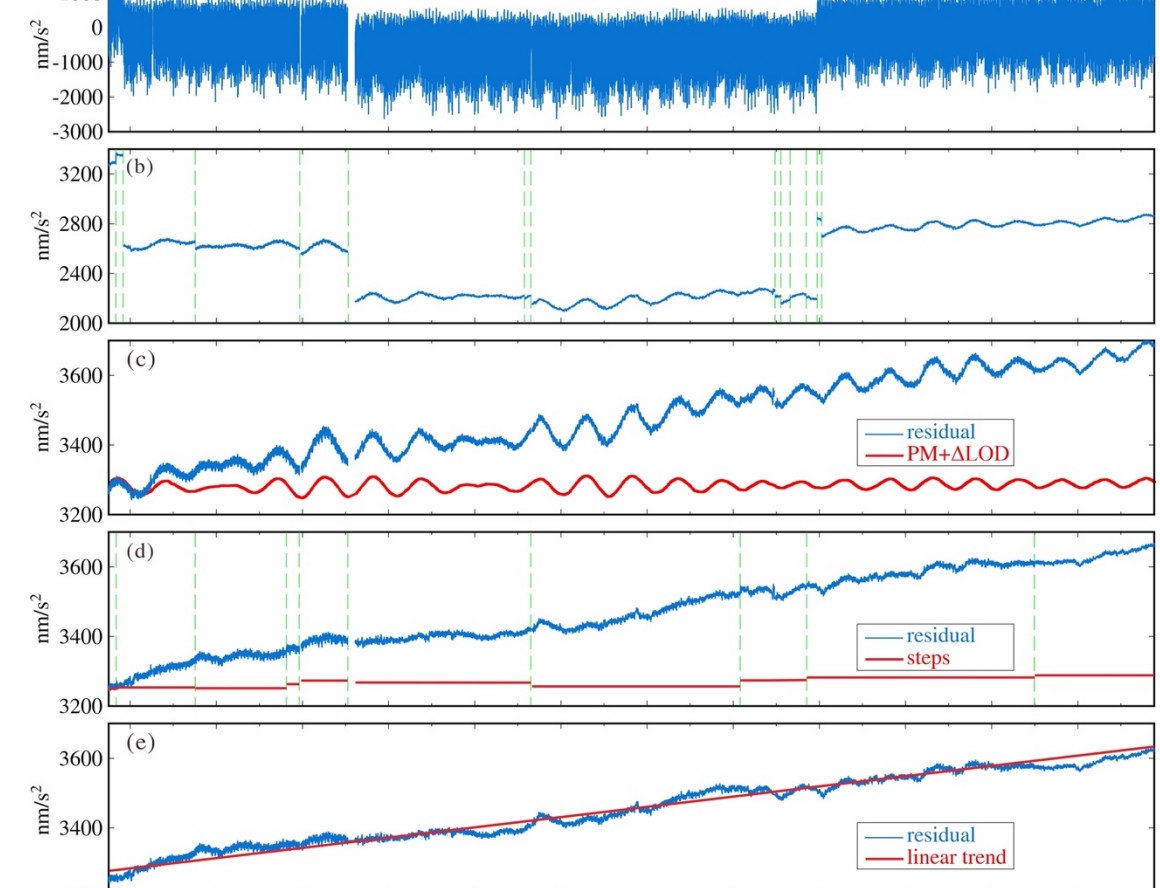

**Figure 1.** SG data pretreatment procedure, taking the CB record as an example. From top to bottom: (**a**) hourly SG record; (**b**) SG residual after removing the data spikes, local synthetic tide, atmospheric loading and non-tidal oceanic loading effects; (**c**) SG residual by adjusting offsets via the PM and ΔLOD time series; (**d**) SG residual after removing PM and ΔLOD effects, and the red lines show the small steps obtained from filtering; (**g**) hourly SG residual after further step corrections and gap filling, together with the linear trend; (**h**) daily SG residual after detrending and decimation.



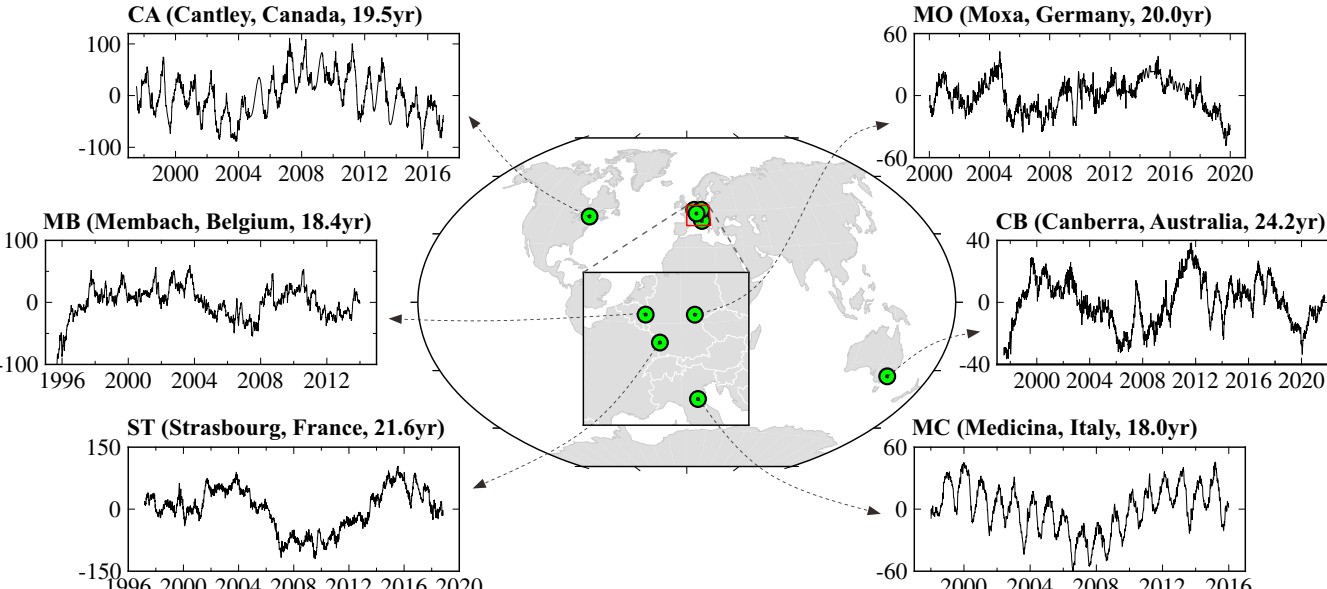

**Figure 2**. The used SG gravity residuals (in nm/s$^2$) obtained at the selected 6 stations. The dots on the map indicate the station locations, and the subfigures show the six residual series.


It should be noted that we did not subtract the hydrological loading effects using the provided models in the above procedure. On the one hand, the Fourier spectra for the modeled atmospheric, PM+ΔLOD, and non-tidal oceanic effects show no obvious 5.9 years peak (Supplement Fig. S4). Though the intradecadal (the 5-10 years period band) frequency bins correspond to some small peaks in the Fourier spectra of the hydrological model for almost all stations, the corresponding AR-$z$ spectra (introduced

in next section) reveal no significant spectral peak around the target frequency bin in (Supplement Fig. S5). On the other hand, we find that the amplitudes of the annual fluctuations in the hydrological models obviously differ from those in the SG residual series in Fig. 2, especially at the MB, MO, and ST stations. Hence, the removal of the hydrological loading effects is very likely to disturb the SYO signal retrieval.

**3 Spectra analysis**

We devote this section to detecting the potential SYO signal and dissecting its interferential signals in the interannual-to-decadal band. The conventional Fourier spectrum, Morlet wavelet spectrum, and stabilized AR-$z$ spectrum (autoregressive spectrum implemented in the complex $z$ domain) are applied here to deal with the SG residual sequences for mutual verification. The first two spectra methods can yield the essential information of a specific signal identified by us. The stabilized AR-$z$

spectrum, highly sensitive to the harmonic signal of decaying sinusoids, is quite efficient to determine the complex frequencies, especially for some adjacent signals that cannot be identified due to low frequency-resolution in the Fourier spectrum (see details in Ding and Chao, 2018b).



The Fourier, AR-*z*, and Morlet wavelet spectra of the residual SG time series at six SG stations in the 0-0.8 cpy frequency band are depicted in Figs. 3 and 4. As noted above, the Fourier spectra can marginally resolve some interannual-to-decadal

signals due to data length limitation. Still, they visibly indicate some well-known periodic signals in the AR-*z* spectra, with their instantaneous information in the Morlet wavelet spectra. The annual variation has quite complex physical mechanisms and is usually removed beforehand by using least-square fitting when studying long-period terms. The well-resolved ~2 years signals can be attributed to the quasi-biennial oscillation (QBO) of the tropical atmosphere with an average period of 26.3 months, characterized by the periodic cycle of stratospheric westerly/easterly phases (Belmint and Dartt, 1968). The most

intractable long-period terms are the ~8.5-18.6 years and ~3-5 years signals, which seriously pollute the potential SYO term and are hardly recognized. The Fourier peaks in the 3-5 years band are well resolved to three periodic signals of ~4.2, ~3.65, and ~3.2 years in the AR-*z* spectra, close to the observed 4.9 years and 3.65 years signals in ΔLOD (Chen et al., 2019), which can largely be accounted for by local or non-local atmospheric/oceanic/hydrological (AOH) sources (e.g., the tropospheric El Nino/Southern Oscillation in the tropical Pacific). Besides, a ~2.6 years signal with different local amplitudes is also clearly

detected to approximate the confirmed 2.5 years signal in PM data in Chen et al. (2019).

For timescales longer than 5.9 years, the residual SG consequences are dominated by quasi-periodic oscillations around 8.5-18.6 years. This range may contain the ~8.5 years, ~11 years, ~13.5 years, and ~18.6 years signals. The limited data length is believed to be insufficient to identify such decadal oscillations, even using the AR-*z* spectrum. Ding (2019) first found an ~8.5

years signal in the 1962-2016 ΔLOD data, and Ding et al. (2021) further investigated its fluctuation characteristics. However, its origin is presently unidentified, and it perhaps has some relevance with the found 4-9.5 years signal in the geomagnetic field from a core flow model (Gillet et al., 2015). The ~11 years signal was clearly shown in the ΔLOD and suggested to stem from the 11 years solar cycle by Currie (1980). Currie (1981) and Le Mouël et al. (2019) further studied the relationship between the ~11 years signal in the ΔLOD and the 11 years solar cycle. Jackson and Mound (2010) also found a ~11.5 years

signal in geomagnetic observations consistent with the solar cycle. Ding and Chao (2018a) and Chao et al. (2020) showed a ~10.5 years signal in the Earth's dynamic oblateness Δ$J_2$ time series. Ding (2019) further distinguished the two clear signals (~11 years and ~13.5 years) from the yearly 1760-2018 long-period ΔLOD data. The ~13.5 years signal corresponds to the ~14 years signal in PM identified by Höpfner (2004), and was suggested to relate to the convective outer core (Kuang et al., 2017). The ~18.6 years signal is a well-known lunar tidal signal (lunar nodes) and has been found in numerous datasets, e.g.,

ΔLOD (Chao et al., 2014; Ding and Chao, 2018a, b; Ding, 2019) and Δ$J_2$ (Cheng and Tapley, 2004; Ding and Chao, 2018b; Chao et al., 2020).

Regarding the target SYO signal, in the Fourier spectra (see the top panels in each subgraph of Fig. 3), there is no helpful information that can disclose the SYO signal from all records. Meanwhile, there are significant peaks for the ~5.9 years signal

in the corresponding AR-*z* spectra (see the bottom panels in each subgraph of Fig. 3); this finding benefits from the high-efficiency discernibility of the AR-*z* spectra for adjacent signals. From the Morlet wavelet spectra in Fig. 4, the ~5.9 years





signal can also be identified, with its instantaneous information showing that the SYO is disturbed slightly by adjacent signals, especially the ~4.2 years and ~8.5 years signals.

**Figure 3**. The Fourier and AR-*z* spectra of the residual SG time series in the secular period band at six selected stations. Red dashed lines denote the reference period 5.9 years of the SYO signal. The gray area denotes the background noise level of the residual SG in the 0-1.1 cpy frequency band, and the vertical shadows and arrows indicate the prominent annual-to-decadal oscillation signals.





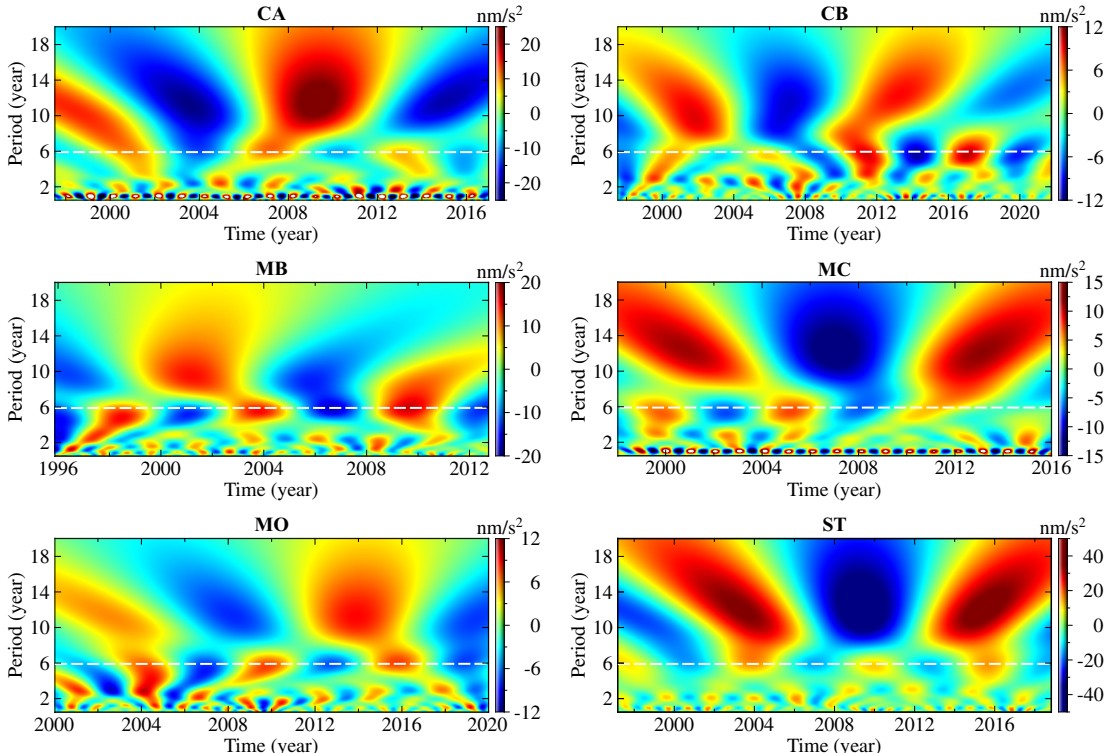

**Figure 4**. The Morlet wavelet spectra of the residual SG time series in the secular period band at six selected SG stations. The white dashed lines denote the reference period 5.9 years of the SYO signal.

## 4 Retrieval of the SYO signals

### 4.1 Determination of the SYOs

For length-limited records, when there are some other periodic signals around about the target low-frequency signal, directly using a least-square fitting process for the target signal cannot obtain its accurately amplitude and phase estimates, hence the surrounding signals should be fitted and removed first. According to the results shown in Sect. 3, we need further remove the non-target long-period signals, as Holme and de Viron (2013) and Ding (2019) have done.

Here we use 8 harmonic signals, i.e., ~18.6/13.5, 8.5, ~4.2, ~3.65, ~3.2, ~2.6, ~2, and ~1 years signals (all of which have been detected or interpreted above), to fit the low-frequency terms of all residual SG time series (referred to as R1) by a least-squares process (Fig. 5). The green curves denote the low-frequency fitting results (F1) based on these 8 signals, and the linear trends are also contained. The fitting and actual fluctuations are in good agreement. The Morlet wavelet spectra for the residuals R2 after the fitting and removing process (i.e., R1-F1 in red) expose prominent and noise-free ~5.9 years signals in the intradecadal period band. Since the simulated tests implemented in Supplement Fig. S6 have verified that such fitting and





removing process would not affect the SYO retrieval, we determine all the analogous spectra signals for R2 as the actual SYO signals in the original time series, which also were confirmed in the AR-*z* spectra. We finally obtain the periods of the SYO signals within the range of 5.84 to 5.92 years, which are consistent with the estimates from ΔLOD, GPS displacement, and geomagnetic data (Mound and Buffett, 2006; Ding and Chao, 2018a; Ding, 2019; Ding et al., 2021; Hsu et al. 2021).


**Figure 5.** The SG residuals after fitting and removing process and their corresponding Morlet wavelet spectra at six selected SG stations. Blue curves denote the residual SG time series R1, Green curves denote the least-square fitting of R1 on behalf of the prominent annual-to-decadal oscillations, and red curves denote the residual R2 (R1-F1). The white dashed lines show the ~5.9 years period.






Due to the short time spans we here regard the gravity SYO as a stable periodic signal, though the SYO in the ΔLOD has been confirmed to be an unstable fluctuation (Chao et al., 2014; Ding et al., 2021). In terms of the R2 sequences, we thus can use the least-square fitting again to recover the gravity SYO time series at all SG stations, as shown in Fig. 6. The gray curves indicate the residual R2 time series, and the red curves indicate their least-square fitting results of the SYO. Combined with

the geographical distribution of SG stations, the ~5.9 years gravity changes in the Western European region (zoom-in area) with the SG stations concentrated vary from 0.5 to 0.9 µGal (1 µGal = 10 nm/s²), i.e., 0.61±0.23 µGal, 0.53±0.12 µGal, 0.86±0.18 µGal, and 0.74±0.21 µGal, respectively corresponding to the amplitudes of the MB, MC, ST, and MO sequences. In addition, the SYO amplitudes at CA and CB are respectively 0.79±0.28 µGal and 0.73±0.14 µGal. In terms of phase, the adjacent three stations MB, ST, and MO have similar phases. The results reflect the ~5.9 years gravity effects in the

corresponding regions on the whole, since the SYO may be interpreted as related to large-scale deep interior dynamics.

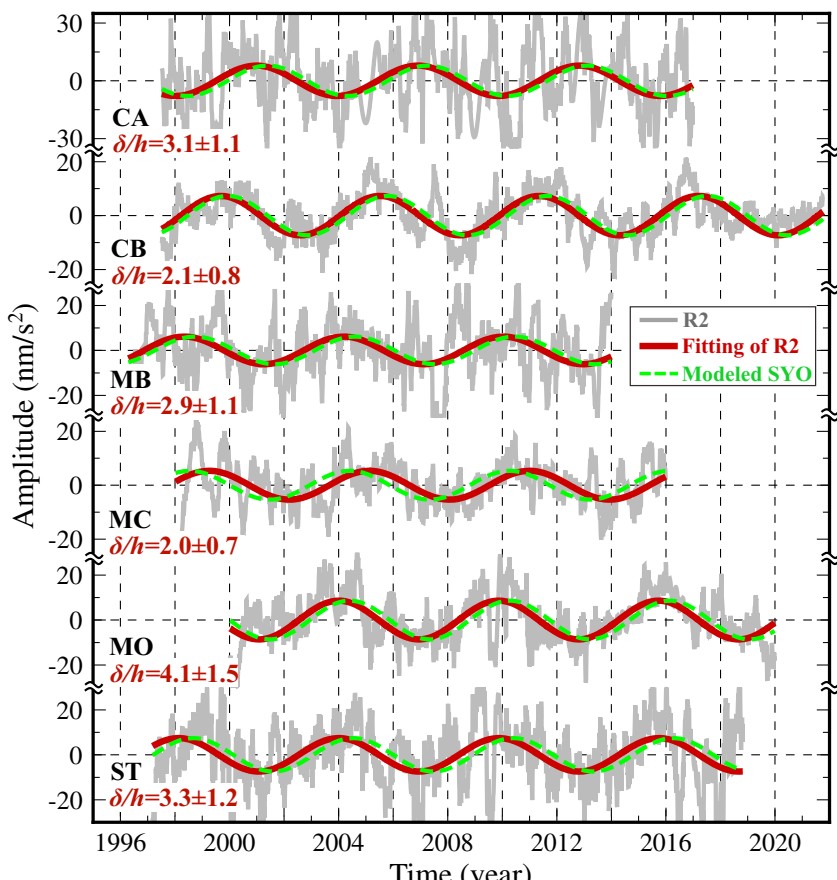

**Figure 6**. The recovered SYO time series from the six selected SG observations. The gray, red, and green curves respectively denote the residual R2 time series, their least-square fitting, and corresponding modeled SYO signal after amplification. The estimated parameters $\boldsymbol{\delta/h}$

are marked on the left for each series.



## 4.2 Comparisons with a derived time-varying gravity model

Ding et al. (2020, referred to as D20) constructed a time-varying 3-D displacement model of the SYO from global GPS displacement observations (see their Eq. 4a). According to Wahr (1985), for a more realistic rotating earth, to the first order,

the vertical deformation $u_{\mathrm{U}}$ and surface gravity $\Delta g$ are proportional to the potential $\Delta V$ in the relations

$$u_{\mathrm{U}} = \frac{h}{g}\Delta V, \Delta g = -\frac{2\delta}{R}\Delta V, \tag{1}$$

the D20's vertical displacement model can thus be written in terms of the gravity changes as

$$\Delta g = -\frac{g}{R} \cdot \frac{\delta}{h} \cdot \frac{a}{2}\sqrt{\frac{15}{2\pi}}\sin^2\theta\cos\left(\frac{2\pi t}{T} + \varphi_1 + 2\lambda\right), \tag{2}$$

where $g$=9.7803278 m/s² is equatorial gravitational acceleration, $R$=6378.1 km is the Earth's mean radius, and $\theta$ and $\lambda$ are the

colatitude and longitude of the used station, respectively. $a$ and $\varphi_1$ are the estimated amplitude and associated initial phase of the complex equivalent "excitation" sequence (see D20's Eq. 1); D20 has given their estimated values $a$=4.37±1.6 mm and $\varphi_1$=1.119±0.03 rad. The period of the SYO applied is set to 5.865±0.024 years by weighted averaging our estimated results. Here we actually do not know the physical mechanism about how the SYO causes the displacement and gravity changes; hence, we consider $\delta/h$ as an unknown parameter named the amplitude transform factors (or amplification factors). If the excitation

is an outside force, similar to the Sun/Moon gravitation, $h$ and $\delta$ are just the vertical displacement and gravity Love numbers, roughly equal to 0.61 and 1.16 respectively (Wahr, 1985; Dehant et al., 1999). Namely, if we obtain $\delta/h$=1.16/0.61≈1.9, then the SYO may be caused by the external source of the Earth; otherwise, it may come from the internal source. In the following, we first simply set $\delta/h$=1, then we can compare the modeled gravity amplitude with the observed gravity amplitude to obtain this parameter.


According to Eq. (2), we can construct the SYO gravity time series for a specified station. The green curves in Fig. 6 indicate the modeled SYO signals for comparisons after amplification, where each subgraph lists the corresponding parameter $\delta/h$. With regard to the phase, the modeled SYO gravity time series could match our recovered results in the time domain, having a maximal deviation within $0.246\pi$ (or 0.72 years time deviation for MC), which accords with the $Y_{22}$-phase pattern of the ~5.9

years oscillation. As for their amplitudes, the SYO gravity model requires that the stations with similar latitudes have approximate fluctuation intensities, which is suitable for the CA, MB, MC, ST, and MO stations. Considering our simulated test has confirmed that the residual offsets may affect the observed ~5.9 years signal for the phase less than $0.1\pi$ and the amplitude less than 0.18 μGal (see Supplement Fig. S3), the further caused errors for the amplification factors are acceptable in the estimated error ranges.



## 5 Discussions and Conclusions

From the Fourier, AR-$z$, and wavelet spectra results for six usable SG residual sequences worldwide, we confirm the presence of the ~5.9 years periodicity in surface gravity observations. Meanwhile, some other long-period signals (including 18.6 years lunar nodes, quasi-biennial oscillation, and ~2.6-4.2 years signals related to the AOH sources) in the interannual-to-decadal periodic band are also identified with periods close to previous results. After removing these long-period terms using a least-square fitting process, we successfully retrieved the SYO gravity time series and obtained a weighted average period of 5.865±0.024 years of the SYO signal. The existence of the ~5.9 years oscillation contributes to surface gravity fluctuations with the amplitudes of ~0.5-0.9, ~0.8, and ~0.7 μGal in Western Europe, Cantley (Canada), and Canberra (Australia), respectively. Comparisons between the recovered SYO time series from surface gravity observations and the time-varying gravity model of the SYO derived from Ding et al. (2020) demonstrate that our results are well consistent with the modeled results in terms of phase. Meanwhile, as for the amplitude transform factor $\delta/h$, we obtain the values of about 2.1 and 2.0, respectively at the Canberra (Australia) and Medicina (Italy) stations, and about 2.9 to 4.1 at the other stations; if taking into account the estimated errors, they are consistent with each other. Given that the estimated parameters $\delta/h$ are generally larger than 1.9 (in the case of the Earth's external source), we may conclude that the corresponding SYO in displacement and gravity observations should come from the Earth's internal source. Although we cannot quantitatively explain the possible physical mechanism, the estimated parameters $\delta/h$ will still help to trace the origin of SYO.

In the previous studies, Mound and Buffett (2003) predicted a 0.06 μGal gravity signal characterized by a $Y_{22}$ surface spherical harmonic for the ~6 years oscillation observed in ΔLOD by developing a theoretical model of the core-mantle system; Dumberry (2010) suggested that typical gravity variations from pressure changes over decade timescales are expected to be of the order of 0.07 μGal for harmonic degrees 2; Ding and Chao (2018a) suggested the surface vertical displacement as 4.3 mm, corresponding to the surface gravity change of about 0.08 μGal, considering the CMB (core-mantle boundary) pressure as a possible ~5.9 years signal mechanism. Although we have used relatively high-quality SG records and tried to make our data pretreatment and fitting process reliable, the remaining magnitude discrepancies between our SYO retrieval results of this study and the previous modeled values are still significant. Since the dynamical core processes could contribute to a quasi-steady part of the non-spherical gravity field at the order of a few hundred nanoGals (Greff-Lefftz et al., 2004), we assume that the Earth's core processes may also be coupled to the motions of other regions in the Earth, and thus form secondary effects; And the magnitudes of which are not necessarily smaller than the direct effects. In addition, Gillet et al. (2020) and Rosat et al. (2021) concluded that the CMB pressure could not cause such significant displacement/gravity changes as observed in Ding and Chao (2018a) (and in this study). Here we do not deny that the mechanism suggested by Ding and Chao (2018a) may be inappropriate, but there are still some doubtful points in Gillet et al. (2020). For example, 1) they used the internal loading Love numbers from Dumberry and Bloxham (2004), which are different from either of those suggested by Fang et al. (1996) and Greff-Lefftz et al. (2004); 2) the dynamical pressure at the surface of the core they used came from the core flow



models which were inverted from geomagnetic observations; however, such core flow models were obtained based on a few assumptions, and different assumptions may obtain different results (e.g., Finlay et al., 2010; Asari and Wardinski, 2012; Gross, 2015; Homle, 2015); 3) the process for inverting the core flow model from the geomagnetic observations involves fitting the fluctuations in the observations with spline curves, while a spline curve may be consisted by different periodic/quasi-periodic oscillations related to different harmonic coefficients; simply spline curve fitting may cause different $Y_{lm}$-related signals to leak into a given $Y_{l'm'}$ term, and therefore obtain inappropriate results and explanations.

Our surface gravity observations reveal that although the origin of surface SYO should be the Earth's core processes, the specific physical mechanism is complex, and the above-mentioned mechanism models are not sufficient to explain the current observations from GPS and surface gravity. Comparing with the torsional oscillation or MICG mechanisms, we tend to suggest that the MAC waves, which arise from the interplay between magnetic, Archimedes, and Coriolis forces (Braginsky, 1993; Buffett et al., 2016; Jaupart and Buffett, 2017), may be a possible source of the SYO. Buffett et al. (2016) indicated that larger fluctuations are possible when electromagnetic stresses couple MAC waves to flow in the interior of the core. However, the detailed mechanism still needs a lot of theoretical derivations and dynamic simulation experiment verifications. Further in-deep tests of the possibility of this explanation with the results from other datasets (especially GPS and geomagnetic observations) are needed too.

Future improvements in gravity observations allow the contribution to surface gravity variations from dynamical core processes to be more reliably detected. Besides, it should be noted that offset correction for gravity data pretreatment is still a challenging problem, just as in other geodetic/geophysical data pretreatment. For SGs, this can only be validly done using repeated measurements from absolute gravimeters to obtain the best time series. However, this work is hard to achieve; hence we have to treat the SG offsets with different amplitudes using specific methods. Moreover, the number of current usable SG records is not enough to verify the $Y_{22}$ spatial pattern of the gravity effects induced by the ~5.9 years oscillation. High-quality SG data accumulation is still to be continued.

*Data availability.* The SG datasets used in this study are from the IGETS database at GFZ (German Research Centre for Geosciences) Potsdam (https://isdc.gfz-potsdam.de/igets-data-base/). The models of surface gravity variations used here are available from the EOST Loading Service (http://loading.u-strasbg.fr/surface_gravity.php).

*Author contributions.* WL conducted the research. HD supervised the research work. WL prepared the manuscript. HD reviewed and edited the paper. All authors contributed to manuscript revision and read and approved the submitted version.

*Competing interests.* The contact author has declared that none of the authors has any competing interests.



*Acknowledgements.* This study is supported by the National Natural Science Foundation of China (Grants: 42204003, 41974022, 42274011, 42274033), the Educational Commission of Hubei Province of China (Grant: 2020CFA109), the Project Supported by the Special Fund of Hubei Luojia Laboratory (Grant: 220100002), and the Open Fund of Wuhan Gravitation and Solid Earth Tides National Observation and Research Station (Grant: WHYWZ202206).

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
