# Peer review of "The possible 5.9 years oscillation identification from superconducting gravimeter observations"

_EGUsphere, 2023_

## Referee Comment (RC2)

General comments

I thank the authors for having responded to nearly all my points and brought some new information related to my comments. However, the new analysis do not bring convincing arguments for a core-origin of this 6-yr oscillation, on the contrary, they demonstrate further the role of surface mass variations, as I explain here after:

**(1) Verification using hydrological loading models at IGETS sites**

Thank you for making the analysis of various hydrological loading models. When looking at the CWT spectra of Fig. 3 in the Reply_comment.pdf, we do see clearly a 6-year oscillation at most SG sites. The scales are not the same than in the paper, in particular the time window is much longer here, giving a less stable impression for this oscillation. We also see from the FFT spectra that the interannual content between different hydrological loading products is quite different. The hydrological models being imprecise, there is no reason to have peaks exactly aligned with the SYO, particularly also, because you performed the analysis on different and longer time windows than for SGs. So we cannot exclude from this analysis the role of hydrology since the models are not precise enough (large dispersion in their interannual content). That confirms also the analysis by Pfeffer et al. (2022, https://doi.org/10.5194/egusphere-2022-1032) that I mentioned in my previous comments.

**(2) Verification using global gridded precipitation data**

Here also, you have performed the wavelet analysis on a much longer time window than for SGs, hence a frequency shifted with respect to the SYO, and less clear. We clearly see on Fig. 6 and Fig. 7 the presence of an oscillation around 6-years. If you consider the same shorter time-window than for SGs, the SYO will be clearer. It is particularly stronger and clearer after the 1990s, when you indeed analyzed the SG time-series…

Consequently, I disagree when you write that you did not find any significant and consistent 5.9-yr oscillation in hydrological loading model and global gridded precipitation data, since in fact we do see it. We would see it better if you use the same time-windows as for SGs.

Line 118-119: I disagree also with the claim that the "hydrology-excited LOD time series does not contain the ~5.9-year signal" since we do see it in Fig. 4(c) of Rosat & Gillet (2023, https://doi.org/10.1016/j.pepi.2023.107053) with a yet small amplitude around 0.012 ms.

**(3) Verification using climate indices, GMST, and GMSL**

Same remark can be made on Fig. 8, where, contrary to what you wrote, we do see a signal around 6 year, which will also be clearer when using smaller time-windows as for SGs.

In Fig. 9 also the time-window is much too long to identify the 6-yr content since we add more spectral content and cannot be compared to the shorter (and less resolved) SG time-series.

In Fig. 10, the trend should be removed before computing the FFT and the CWT else you introduce large long-period content. The time-window used is here also much too long to be compared with SG time-series.

**(4) Verification using the oblateness ∆J₂**

Only a certain distribution of terrestrial water storage variations would result in ∆J2 since ∆J2 is related to zonal degree-2 pattern (C20 in terms of Stokes coefficient). The hydrological content at interannual time-scales is mostly related to degree-2 order-2 geographical pattern (C22 and S22 in terms of Stokes coefficient; see for instance Meyssignac et al. 2013). Interannual variations in degree-2 Earth's gravity coefficients C2,0, C2,2, and S2,2 reveal large-scale mass transfers of climatic origin, Geophys. Res. Lett., 40, 1-6.), so of course it would not contribute much to ∆J2. This argument is then inappropriate.

About the AR-z spectrum

I thank the authors for sharing the information about the availability of the AR-z spectrum method. Since "the power of the peaks in the AR-z spectrum directly correlates with their stability rather than their actual amplitude", I would be curious to see how it performs on the hydrological loading time-series (same time-windows than SGs) that you analyzed by FFT but for some better hydrological models like ERA5_land. The ERA-interim model that you used in the Supplementary material is of poor quality (as seen by comparison with GRACE for instance).

Concerning the ratio δ/h for continental hydrology

You have not responded to my remark concerning the influence of **local contribution** of hydrological mass changes that play an important role on the value of this ratio.

Lines 274-275-277: Again, I disagree in the fact that hydrological loading has a negligible influence (see previous comments and wavelet spectra that exhibit 6-yr content, particularly during the time-periods on which SG data were analyzed).

Concerning SG data

I have downloaded Level-2 SG time-series from IGETS website and performed the CWT and FFT analyses of all time-records longer than 18 years. The step corrections as well as the instrumental drift correction are very sensitive processes that would modify consequently the interannual spectral content. For a few stations, the SYO is clearly conspicuous but for others it is not. It really depends on the station considered and the pre-processing of data. Even among the stations you have used, I have quite different spectra. I do see it clearly for Strasbourg and Metsahovi (with a nice anti-correlation with hydrological loading), but not for Canberra, Cantley, Medicina, Membach, Moxa (not at 6-yr but different periods, like 4-yr, 5-yr or 7-yr). If the SYO was so stable, a different pre-processing of same datasets (with a weak proportion of steps and gaps) should still make it visible when the time-series is long enough.

---

## Referee Comment (RC3)

General comments

I thank the authors for having responded to my points, some of my points were misunderstood other I still disagree:

*(1) Previous evidences from different observation data (e.g., ΔLOD, geomagnetic field, GNSS, and polar motion) have indicated that the SYO in ΔLOD is an almost stable oscillation with attenuation larger than 180.*

I agree that the SYO is clearly observed in ΔLOD, it does not mean a SYO in gravity (associated with mass variations mostly but to a lesser extent to centrifugal effects) would be the same signal, in particular here we are looking at mass contributions. A large part could (I think it is most probable) result from surface mass redistribution. It is possible that a core origin SYO is also present, but the problem I mentioned is how to disentangle it from surface mass redistributions, like hydrology and climatic events, which has large interannual content as demonstrated by the previous studies I mentioned.

*(2) Regarding the verifications using global gridded precipitation data, climate indices, GMST, and GMSL, you both posed questions similar to that in validation using hydrological models. Specifically, all of these validations should be conducted using smaller time windows as for SGs. It is imperative to reiterate that it is necessary to use observations for extended durations, preferably in alignment with the ΔLOD time, to ascertain the presence of a stable ~5.9-year fluctuation before establishing a correlation between it with the SYO in LOD.*

I agree that it is necessary to have extended durations to validate the presence of a SYO, but in the SG data you do not have the required extent (contrary to gridded precipitation data, climate indices, GMST, and GMSL). But for a fair comparison with your shorter SG data, you should compare precipitation data, climate indices, GMST, and GMSL on the same time windows.

*However, it must be founded on a precondition that the SYO is actually present in the hydrological models.*

No, there is no need for any precondition about the presence of a SYO, since it is well-known that there are interannual climatic events. As for a precondition that the SYO is actually present in SG data is rather your own hypothesis since when applying the AR-z spectrum we cannot see it and for some SG sites I do not see it either with the FFT. I have downloaded the AR-z code and made synthetic tests using values to mimic the SYO (55 years of data sampled at 0.1 yr, a SYO of amplitude 0.2 buried in a white noise of 0.3) (Fig. (a)).

```
Fs = 10;                % Sampling frequency
T = 1/Fs;                % Sampling period
L = 55*Fs;                % Length of signal
t = (0:L-1)*T;          % Time vector

% construct the signal data
B = 0.2;
S = B*sin(2*pi*1/6*t);
% add the noise data
X = S + 0.3*randn(size(t));
```

(a)

[Figure]

(b)

[Figure]

(c)

[Figure]

(d)

[Figure]

When there is noise (even just white noise not colored one) besides the SYO, the FFT clearly shows the spectral peak Fig.(b), but with the AR-z spectrum, (Fig.(c) and (d)) it is difficult to obtain the correct peak, even with different Q-values, since many spurious peaks appear. When the random noise has changed, the AR-z spectrum is also quite different and the peaks are not same. With the stable version, I assume you will need to adapt weights computed as noise amplitude in different frequency bands, but here also you need a strong *a priori* since you aim at enhancing a peak around 6 yr.

(3) *Observing Fig. 4(c) in Rosat & Gillet (2023), an extremely weak 5.9-year signal with an amplitude of ~0.012 ms is present in the wavelet transform coefficient spectrum of the HAM data. Nevertheless, it can be observed that the amplitude is nearly congruent with the background noise level of ΔLOD. As depicted in Fig. 1, the background noise level of ΔLOD after removing the AAM and OAM effects is ~0.01 ms. It also can be seen in Fig. 1d of Ding (2019, EPSL) and Fig. 1b of Ding et al. (2021, JGR). That is to say, despite the presence of a ~5.9-year signal in the hydrology-excited LOD, the associated peak is likely to be buried in the background noise (the SNR is very low). Statistically, the signal is unreliable.*

I agree that the observed SYO in HAM in that paper is very small, but their statistical levels on Fig. 4(c) still detect it. Your synthetic test is something I did also, and when using a colored red noise, it is even worse. This noise effect could also be what you observe in SG data. Only statistical significance tests could help to decide whether you see noise or not, particularly in your case, when SG time series are not long enough. As for your FFT spectra on Fig. 2 in your reply 2, they are not to-scale, since they also should exhibit the fake peaks as seen in the CWT. A proper scaling would show it. That is why it is also important that you put statistical levels in your AR-z spectra and CWT. By the way, as concerning SG data, using a colored noise model would be more appropriate than a white noise at such long periods, since SG noise levels (represented by power spectral densities) increase with periods.

*it may be more appropriate to rephrase the statement as follows: The HAM effect made very small contributions to the intradecadal period band in the ΔLOD.*

I agree with that sentence but it is written in their paper:" HAM excitation function has a small but non-negligible contribution to the interannual LOD fluctuations."

*(4) It can be affirmed that the terrestrial water storage (TWS) variations can result in the C20 variations, which are directly reflected in ΔLOD. Besides, it is worth noting that Chao and Eanes (1995) demonstrated global gravitational changes due to atmospheric mass redistribution, which can also manifest in the C20 variations. Therefore, if an SYO in the TWS variations is observed, it must appear in ΔJ2, and further will be reflected in ΔLOD. However, in our last reply, it has been determined that there is no observable SYO signal present within ΔJ2. Given the stable SYO signal present in ΔLOD, the only explanation is that the SYO signal in ΔLOD does not have any correlation with the TWS variations.*

"The hydrological content at interannual time-scales is mostly related to degree-2 order-2 geographical pattern". Sorry, I should have been clearer in my sentence (use "most important" instead of "mostly"). Meyssignac et al. (2013) have shown indeed that variations in land water storage are more important on C20, but they also showed that there are large water mass redistributions over the oceans, particularly due to climatic events, in longitude (Y22 pattern), and it is not negligible on S22. So restricting hydrology signal to land water storage in a Y20 pattern results in an approximation that miss a part of water mass signal. Since SG gravity measurements are very sensitive to mass changes, they will record it too. Using J2, which is related to C20 and Y20 pattern, as an argument for the absence of the SYO in hydrological data is hence invalid.

*Besides, you claimed that "Only a certain distribution of terrestrial water storage variations would result in ΔJ2." It has no basis at all. According to Chao et al. (2020), "J2 is the (normalized) zonal quadrupole of the Earth's density." "The zonal [degree-2, order-0] component of any mass redistribution will contribute to the time variation ΔJ2.*

**This is exactly what I said**! "*Only a certain distribution of terrestrial water storage variations would result in ΔJ2*" that is the Y20 spherical harmonic pattern. The Y22 mass redistribution does not affect C20/J2, but hydrology has a non-negligible Y22 component (Meyssignac et al. 2013).

*(5) The revised manuscript will present the results of implementing AR-z into some better hydrological models like ERA5_land.*

Ok, thank you.

*(6) As demonstrated in our last reply, both current global hydrological models and global precipitation data do not indicate the presence of a stable and consecutive ~5.9-year oscillation*

Well, not with SG data either since the time windows used are too short with respect to the ones you used for hydrological models and precipitation data. And your synthetic tests with white noise in Fig. 2 of your reply 2 show it could be an artefact of noise too... Statistical confidence levels would be necessary.

---

## Referee Comment (RC4)

**Review to the Article 'The possible 5.9 years oscillation identification from superconducting gravimeter observations' by Wei Luan and Hao Ding**

June 19, 2023

**1   General comments**

The paper raises the interesting question if a 5.9 year oscillation (SOY) observed in other geodetic time series, can also be detected in SG data. I think this is a relevant question and it might help to interpret the origin of such an oscillation.

The authors come to the conclusion, that they can detect SOY signals in the data and estimate their amplitudes and phases.

However, the preprocessing of the data, especially the correction of instrumental disturbances, is very critical, when analysing very long time series of SG data. Especially, for the step correction I have doubts if the methods used in this paper are completely valid. Some of the methods also need more explanation. Further, the accuracy of these corrections might be overestimated by the authors, which highly affects the detection and the interpretation of the 5.9 year oscillation with the small amplitude of $5\,\mathrm{nm/s^2}$ to $9\,\mathrm{nm/s^2}$.

Further, as highly discussed between the authors and Reviewer 1, the origin of such an oscillation might still be debatable. As I see my expertise rather in the processing of SG data, I don't want to add many comments to this discussion.

**2   Specific comments**

**2.1   Data source**

I would like to comment on the confusion about the "h2" database of IGETS.

For most of the stations in the IGETS database three different kinds of Level 2 data exist. They are specified by the code in the ending of their filename: `22.gpp`, `32.gpp` and `h2.gpp`. The meaning of the code "22" and "h2" is documented by Voigt et al. (2016). The procedure to produce the "32"-data is explained by Boy et al. (2023), however the code "32" is only mentioned by Boy (2022).

The "22"-files should contain data with a sampling interval of 1 min, where gaps and disturbances are filled with synthetic data and offsets are adjusted, but it seems (see below) that this is at least not true for big gaps and big offsets. The "h2" files contain hourly sampled data, but Voigt et al. (2016) do not specify which kind of preprocessing was performed. The comparison below indicates that it is the same as for the "22" files.

The preprocessing for "22" and "h2" data is done at UFP (University of French Polynesia).

The "32" files are produced by EOST (Ecole et Observatoire des Sciences de la Terre). They contain data with a sampling interval of 1 min, where gaps are filled and offsets are adjusted.

To clarify, I provide a comparison of the calibrated Level 1 data and all kinds of Level 2 data for the year of 2006 and the gravimeter SG026 at Strasbourg. It is shown in Figure 1. It can be seen that the data spikes in November were removed in all versions of level 2 data. All gaps were filled and steps were adjusted in the "32"-data, while this is not the case for "22" and "h2" data. From visual inspection the "h2" seems to be a downsampled version of "22" data.

As you use your own method to adjust steps, I think it is reasonable that you use a data set, where steps were not adjusted before, which might be the case for "h2" data although it is stated differently by Voigt et al. (2016). In line 86 of your manuscript you mention that you also removed spikes. Does this mean that there were remaining spikes in the "h2" data that you removed on your own? Please clarify on this. If you anyway correct spikes and steps on your own, I think the safest option would be to use the level 1 data. Then you would know that no preprocessing was done and an uncertainty about the preprocessing done by IGETS cannot influence your results.

Please clarify in your manuscript which datasets from IGETS you use (level and code) and which preprocessing steps where done by IGETS.

**2.2 Step correction**

I have some questions and comments on your step detection and step size determination process. First, I consider it a new and interesting idea to estimate the step sizes by fitting the data to the polar motion times series. I have not seen this approach before. However, I have some doubts

[Figure]

Figure 1: **Comparison of Level 1 and Level 2 IGETS data.** The figure shows data of the SG026 at Strasbourg for the year 2006. In the top panel calibrated level 1 data can be found. The other panels show the three different kinds of level 2 data.

if it is completely valid. If I understand it correctly, your fitting process can be described as

$$g(t) - PM(t) - LOD(t) = A \cdot \begin{pmatrix} s_1 \\ s_2 \\ . \\ . \\ . \\ s_N \\ a \end{pmatrix} + \epsilon(t), \tag{1}$$

where $g(t)$ contains your preprocessed data (after removing spikes, synthetic tides, atmospheric and non-tidal ocean loading) and $PM(t)$ and $LOD(t)$ are the models for the polar motion signal and the length of day signal. $s_1...s_N$ are the steps size you estimate, $a$ is a possible linear trend in the data and $\epsilon$ contains all kinds of uncertainties and remaining signals in the data. The design matrix $A$ is

$$A = \begin{pmatrix} H(t - t_1) \\ H(t - t_2) \\ . \\ . \\ . \\ H(t - t_N) \\ t \end{pmatrix}^T \qquad \text{with} \qquad H(t) = \begin{cases} 0 & t < 0 \\ 1 & t \geq 0 \end{cases}$$

where $t_1...t_N$ are the times of the steps.

In this approach $\epsilon$ contains all the periodic gravity signals $g_p(t)$ that you analyse afterwards, the stochastic part of the gravity signal $g_s(t)$, the remaining small steps, $s(t)$ the uncertainty of the polar motion model $\epsilon_{\text{model}}$, and the noise of the SG $\epsilon_{\text{SG}}$:

$$\epsilon(t) = g_p(t) + g_s(t) + s(t) + \epsilon_{\text{model}} + \epsilon_{\text{SG}}.$$

In adjustment theory, minimising the square of $\epsilon$ in Equation 1, is based on the assumption that $\epsilon(t)$ is stochastic with zero mean (Teunissen 2003). This might be true for $g_s(t)$, $\epsilon_{\text{model}}$, and $\epsilon_{\text{SG}}$, but the remaining small steps do probably not have zero mean and the periodic signals in the gravity residuals are not stochastic.

Therefore, I would propose to estimate the size of the steps, and all amplitudes and phases of the periodic signals at the same time. Otherwise you have to justify, why your method is still valid, or, if it is not completely valid, how big the errors are introduced by using it anyway.

Further, please also explain the method in more detail in your paper. As you write in the end

of the paper, the best would be to also include absolute gravity measurements in the step size and drift estimation.

For the smaller steps you use a second method to find and remove them, which is explained in the supplementary material. From there it is very clear what you did to get to the green curve in Figure S2(c). However, please explain how you determine the step times and sizes from the green curve. This is not clear.

Especially, for the bigger steps in SG data a problem in estimating their size is that they often occur together with data gaps and that sometimes a running-in behaviour of the data occurs after the step. How is the accuracy of your step removing procedures influenced by this problem?

Beside the question if and how hydrology needs to be removed, the step correction is the biggest uncertainty in your data processing. Therefore, I appreciate that you analyse the influence of errors in the step corrections on your results. However, how do you come to the conclusion, that continuous downward steps cause the maximal amplitude deviation? I did a synthetic test similar to the one you present in the supplementary material, but for a mixture of upward and downward steps: I simulate a synthetic time series containing white noise with a standard deviation of $10\,\mathrm{nm/s^2}$ (Figure 2(a)), six steps of $4\,\mathrm{nm/s^2}$, which are upward and downward (Figure 2(b)) and a SOY signal with a period of 5.9 years and an amplitude of $6\,\mathrm{nm/s^2}$ (Figure 2(c)). To simplify the synthetic test, I only estimated the amplitude of the SOY in the synthetic dataset with a linear least squares estimation and assumed the phase to be known. The simulated time series, the input SYO signal and the estimated SYO signal are shown in Figure 2(c). The estimated amplitude of SYO in the synthetic data is $8.1\,\mathrm{nm/s^2}$, which means a deviation of more than $2\,\mathrm{nm/s^2}$ from the input value. It is possible that the deviation is even bigger for other configurations. This test should just show that the deviation can be bigger than $1.8\,\mathrm{nm/s^2}$. Furthermore, your synthetic test is only meaningful, if you can be sure that errors in your step determination and step correction process are not bigger than $4\,\mathrm{nm/s^2}$. How did you come to this estimate?

In conclusion, I think the accuracy of the step correction might be overestimated. This leads to an underestimation of the errors of the SOY signal's amplitudes and phases, which finally affects the interpretation of the SOY signal.

**2.3 Spectra**

You emphasise the better frequency resolution of the AR-z spectra compared to Fourier spectra. I agree that the frequency resolution of the Fourier spectra of $\frac{1}{T} \approx 0.5\,\mathrm{cpy}$ is not enough to resolve signals in the 3-5 year band, the 8.5-18.6 year band or the SYO. Is it possible to quantify the frequency resolution of the AR-z spectra. If yes, please do so. This would help to know the accuracy of your determined frequencies.

[Figure]

Figure 2: **Synthetic test for the effect of small steps on the SYO signal retrieval.** (a) shows the simulated noise series, (b) shows the input steps and (c) shows the simulated SOY, the simulated time series and the recovered SOY.

Did you use any taper to compute the Fourier spectra? If yes, please specify.

**2.4   Retrieval of the SYO signal**

How do you finally obtain the periods of the SOY ranging from 5.84 to 5.92 years? (Line 188-189). Do you read them from the Morlet wavelet spectra or from the AR-z spectra? In Figure 5 you give uncertainties for these periods. How did you obtain them? Please specify.

For the same reasons as disused in Section 2.2, I think the estimates for amplitudes and phases of the SOY signal would be more stable, if you estimate them together with the amplitudes and phases of all the other periodic signals.

**2.5   18.6 year period**

In the conclusion you claim you have identified an 18.6 year period in the SG data. However, in the Section 'Spectral analysis' you state correctly that the data length is too short to identify this oscillation.

**2.6   Origin of the SYO**

It is not clear to me, why you think the SYO should originate from the Earth core. Even if you can exclude external and loading sources, why do you think the SOY more likely originates from the core than for example from the mantle?

**3   Technical corrections**

Line 32 -46: A non-expert to core dynamics would have big difficulties to understand the basic ideas of this paragraph. In my opinion this would be fine for a paragraph in the discussion part. For an introduction, however, I think the basic ideas should also be understood by non expert readers. So it would be helpful if you could rewrite this paragraph in a less technical way.

Line 49: (PM, Ding et al. 2019, 2021; Chen et al. 2019) → (PM) (Ding et al. 2019, 2021; Chen et al. 2019)
The same applies to line 51.

Line 181: You write that you are fitting 8 harmonics: $\sim$ 18.6/13.5, $\sim$ 8.5, $\sim$ 4.2, $\sim$ 3.65, $\sim$ 3.2, $\sim$ 2.6, $\sim$ 2 and $\sim$ 1 year. What does '$\sim$' mean in this context? As I understand, you estimate the amplitudes and the frequencies by the fitting procedure, while the frequencies are

fixed? If this is the case, which are the exact frequencies you are using? What do you mean by 18.6/13.5? Do you use both frequencies? Is it one frequency for each station?

Line 259: When talking about pressure changes, are these air pressure changes or pressure changes inside the Earth? Please clarify.

**References**

Boy, J.-P. 2022. *Description of the Level 2 and Level 3 IGETS data produced by EOST (version 2).* `https://isdc.gfz-potsdam.de/fileadmin/isdc/docs/EOSTproducts_v2.pdf`.

Boy, J.-P., J.-P. Barriot, C. Förste, C. Voigt, and H. Wziontek. 2023. "Achievements of the First 4 Years of the International Geodynamics and Earth Tide Service (IGETS) 2015–2019." In *Beyond 100: The Next Century in Geodesy,* edited by J. T. Freymueller and L. Sánchez, 107–112. Cham: Springer International Publishing. ISBN: 978-3-031-09857-4.

Teunissen, P. 2003. *Adjustment theory: an introduction.* Delft: Delft University Press. ISBN: 90-407-1974-8.

Voigt, C., C. Förste, H. Wziontek, D. Crossley, B. Meurers, V. Pálinkáš, J. Hinderer, J.-P. Boy, J.-P. Barriot, and H. Sun. 2016. *Report on the Data Base of the International Geodynamics and Earth Tide Service (IGETS.* Scientific Technical Report STR – Data 16/08. Potsdam: GFZ German Research Centre for Geosciences. doi:`10.2312/GFZ.b103-16087`.

---

## Author Comment (AC1)

**Synthesis test 1:**

$$x = x1 + x2 = 1 \times \cos(2 \times \pi \times \frac{1}{27.2122} \times t + \pi) + 1.5 \times \cos(2 \times \pi \times \frac{1}{27.5545} \times t)$$

[Figure]

**Figure 1.** Synthesis test for the beating period 6yr by superimposing two cosine oscillations with the periods of 27.2122d (lunar nodal month) and 27.5545d (Mm tide), respectively. The data interval is one day, and the data length is 18 years.

**Synthesis test 2:**

$$x = x1 + x2 = 2 \times \cos(2 \times \pi \times \frac{1}{365.25} \times t) + 2 \times \cos(2 \times \pi \times \frac{1}{433} \times t + \pi)$$

[Figure]

**Figure 2.** Synthesis test for the beating period 6.4yr by superimposing two cosine oscillations with the periods of 365.25d (annual wobble) and 433d (Chandler wobble), respectively. The data interval is one day, and the data length is 18 years.

---

## Author Comment (AC2)

General comments

This paper shows that a 6-year oscillation is visible in the time-varying gravity recorded by 6

Superconducting Gravimeters (SGs). The origin of this 6-year oscillation also seen in other geophysical and geodetic time-series is still debated today. A core origin has been suggested but surficial climatic events could also be responsible for that periodic oscillation. This paper completes the catalog of observables containing a 6-year oscillation. It confirms that it is a global effect. They then try to prove that it is of internal origin, but as discussed here after, it is not so clear, and hydrology still prevails. The methods they employed are not new either, and one of their method (AR-z spectrum)

could even raise some criticism. There are also a few scientific flaws and worries that need to be considered and corrected, in particular with respect to the published literature.

**Response**: We are grateful for your comments and corrections on some of the interpretations that we made as well as some of the technical errors that we made, which will assist to improve the work. Our replies to your comments can be found below.

Specific comments (individual scientific questions/issues)

This paper is a follower of a series of studies by Pr. Hao Ding and co-workers who support a core origin for the 6-year oscillation observed in geodetic and gravimetric data, despite some evidence for a more probable surficial origin by Rosat et al. (2021). The criticism can be reproduced here, since hydrological loading does also contain a 6-yr oscillation contrary to what the authors claim (Fig. S5 is not at the same scale as Fig.3 so it is misleading). If you do plot the FFT spectra of ERAin or ERA5

(or ERA5_land) hydrological loading products, as I did, you will see a non-negligible contribution around 6-year. You can also plot the time-series of SG gravity residuals with respect to hydrological loading, band-pass filtered them around 6-yr, and you will see a good correlation between both time- series, for most worldwide SG stations of sufficient data length, but with a time-shift for some stations.

You have to consider cautiously the sign of the local contribution of hydrological loading for underground stations like Moxa, Membach and Strasbourg. Indeed, another group of researchers has shown that the continental hydrology as well as other climatic time-series exhibit a 6-year oscillation e.g. Pfeffer et al. (2022, https://doi.org/10.5194/egusphere-2022-1032), Cazenave et al. (2023, https://doi.org/10.5194/egusphere-2023-312) and Pfeffer et al. (2023, http://ssrn.com/abstract=4388237). This hydrological signal contributes to the observed 6-yr gravity change and would mask any potential signal originating from the core. Consequently, as long as you do not correctly deconvolve gravity data from this hydrological signal, you cannot interpret the 6-yr oscillation as a signal of core-origin.

**Response**: We appreciate your comments and the references that you supplied. Here the main question of controversy is whether or not the hydrological loading contains the 5.9-year oscillation (SYO) (alternatively, whether or not the hydrological effects contribute to the observed SYO gravity change). In addition, you brought to my attention that previous research found that the other climatic time series indicate a 6-year oscillation. In response to these two issues, the following will give detailed verifications using publicly accessible data from four perspectives.

**(1) Verification using hydrological loading models at IGEST sites**

We collected the hydrological loading data at six SG stations selected in this study from 6 different global hydrological models (Fig. 1), including ERA5, ERAin, ERA5-land, GRACE, GLDAS2, and MERRA2, provided by the EOST Loading Service (http://loading.u-strasbg.fr/sg_hydro.php). We use Fourier and Morlet wavelet spectra to verify whether the modeled hydrological loading data contain the SYO signals.

Fig. 2 shows the Fourier amplitude spectra of the modeled hydrological time series at six SG stations. The spectral analysis findings indicate that none of the hydrological models exhibit any noteworthy peaks that align precisely with the SYO frequency (1/5.9years=0.1695cpy, as denoted by the horizontal red lines). Nonetheless, there exist proximate peaks within the period band around 5.9 years, such as the peaks of ~5.4 years at the CB station and ~6.4 years at the CA station. Moreover, we plot the Morlet wavelet spectra of the hydrological data obtained from the ERA5 model and GRACE iterated global mascons in Figs. 3 and 4, respectively. The ERA5 hydrological data at all SG stations exhibit a deficiency in power at the ~5.9-year intradecadal variability. The GRACE hydrologic data at the CB, MO, MB, and ST stations exhibit some degree of power during certain time intervals. However, these signals do not demonstrate significant and consistent SYO patterns in the studied time spans.

[Figure]

**Figure 1. Modeled hydrological loading data series at six SG stations**: (a) ERA5, 1979-2023; (b) ERAin, 1979-2019; (c) ERA5-land, 1985-2023; (d) GRACE, 2002-2022; (e) GLDAS2, 2000-2022; (f) MERRA2, 1980-2023.

[Figure]

**Figure 2.** Fourier amplitude spectra of the hydrological loading data series in Fig. 1. The vertical red dashed lines denote the reference period 5.9 years of the SYO signal.

[Figure]

**Figure 3.** Morlet wavelet spectra of the hydrological data series estimated from ERA5 model.

[Figure]

**Figure 4.** Morlet wavelet spectra of the hydrological data series estimated from GRACE iterated global
mascons.

**(2) Verification using global gridded precipitation data**

According to Pfeffer et al. (2022, 2023), the 6-year oscillation could potentially originate from either precipitation or terrestrial water storage (TWS). Pfeffer et al. (2023) applied a band-pass filter to isolate the frequency band around the 6-year signal from the time series of precipitation or TWS anomalies, derived from satellite gravity observations, in-situ and satellite-based precipitation records, and predictions from global hydrological models. Here we employ the Fourier and wavelet spectra to examine the monthly precipitation data series obtained from the ERA5-Land global gridded precipitation model (https://cds.climate.copernicus.eu/cdsapp#!/dataset/reanalysis-era5-land-monthly -means) and GPCC global gridded precipitation dataset (https://psl.noaa.gov/data/gridded/data. gpcc.html). With the exception of the entirety of the Earth's land surface, our attention is directed towards specific regions (R1, R2, R3) that encompass the used SG stations in order to involve the precipitation effects on both global and regional scales (see Fig. 5).

[Figure]

**Figure 5. Global gridded precipitation data:** (a) The ERA5-Land monthly averaged precipitation model resampled at a 1°×1° grid, in Apr. 2023; (b) The GPCC monthly total precipitation observations sampled at a 2.5°×2.5° grid, in Dec. 2013. The black frames labeled by R1, R2, and R3 indicate the regions covering the used SG stations, respectively CA, MB/MC/MO/ST, CB (green circles).

In Fig. 6 the Fourier and wavelet spectra of the ERA5-Land precipitation data in the global, R1, R2, and R3 regions indicate no oscillation signal at the 5.9-year period. Within the period band around 5.9 years, a peak of approximately 5.0 years is observed, with a low signal-to-noise ratio (SNR) and the amplitude lower than the 95% confidence level (CL) in the frequency band under investigation (Fig. 6e-h). It is worth noting that the wavelet spectra for the global region exhibit evident annual signal, 11-year fluctuation, and 18.6-year lunar tide on annual-to-decadal timescales (Fig. 6a, e). In Fig. 7 an analogous examination is conducted for GPCC precipitation observations based global station data. The Fourier spectra exhibit peaks of ~5.2 years in the vicinity of a 5.9-year period (Fig. 7e-h); however, these peaks do not manifest as consistent oscillatory signals in the wavelet spectra (Fig. 7a-d). Despite the presence of consistent power levels lasting ~5.9 years in narrow time intervals (Fig. 7a), we think that these occurrences may be attributed to errors in modeling or reanalysis.

[Figure]

**Figure 6. ERA5-Land precipitation model:** (a-d) The Morlet wavelet spectra of the average series of the detrended precipitation data in the global, R1, R2, and R3 regions. (e-h) The mean Fourier amplitude spectra of the detrended precipitation data series in the global, R1, R2, and R3 regions. The precipitation time series span from Jan. 1960 to Apr. 2023. The horizontal and vertical red dashed lines denote the reference period 5.9 years of the SYO signal, and the horizontal black dashed lines in (e-h) show the 95% confidence level (CL).

[Figure]

**Figure 7. GPCC precipitation observations:** (a-d) The Morlet wavelet spectra of the average series of the detrended precipitation data series in the global, R1, R2, and R3 regions. (e-h) The mean Fourier amplitude spectra of the detrended precipitation data series in the global, R1, R2, and R3 regions. The precipitation time series span from JAN 1960 to DEC 2013.

To sum up, we do not find any significant and consistent ~5.9-year oscillation in the hydrological loading model data and global gridded precipitation data. From previous studies of ΔLOD, geomagnetic fields, and SLR, it has been observed that the SYO behaves a relatively stable fluctuation with `(e.g., Liao and Greiner-Mai, 1999; Gillet et al., 2010; Ding & Chao, 2018, EPSL; Ding, 2019; Duan & Huang, 2020; Chao & Yu, 2021; Ding et al., 2021). Besides, the hydrology-excited LOD time series does not contain the ~5.9-year signal and the overall influences of the hydrology on ΔLOD are very small, but the atmosphere-excited LOD time series contains a ~5-year signal (Zotov et al., 2020; Ding et al., 2021). Based on the aforementioned verifications and analysis, we preliminarily consider that the hydrological loading does not contain the SYO signal. However, the accuracy of the used reanalysis models or datasets cannot be ascertained with certainty, thus further demonstration is required to determine whether there is a SYO signal in continental hydrology. The optimal and persuasive approach entails showcasing in-situ surface observation data, such as precipitation, soil water, and groundwater level.

**(3) Verification using climate indices, GMST, and GMSL**

The research conducted by Moreira et al. (2019) and Pfeffer et al. (2023) is what we will be referring to in the present verification. Moreira et al. (2019) focused on the interannual variabilities in global mean sea level (GMSL) over 1993-2019, which were linked to various climate modes or indices. Pfeffer et al. (2023) employed a bandpass filter to examine the time series of the global mean sea temperature (GMST) and GMSL over 1993-2002. Here we collected more abundant climate indices from NOAA Physical Sciences Laboratory (NOAA PSL, https://psl.noaa.gov/data/climateindices/), and GMST and GMSL data series from various institutions or workgroups. The classical Fourier and wavelet spectra are still employed to analyze the data to mitigate potential methodological errors.

Fig. 8 displays the Morlet wavelet and Fourier spectra of the monthly climate index series, encompassing PDO, AO, NAO, SOI, AMO, NINO3, AAO, ESPI, and MEI, from 1951 to the present. Just looking at the period band around 5.9 years in the Fourier spectra (Fig. 8j-r), it is evident that, except for AMO, there are peaks of ~5-5.6 years with the amplitudes surpassing the 95% CLs for the majority of climate modes. Upon further observing the wavelet spectra (Fig. 8a-i), no substantial or stable oscillation signal with a period of ~5.9 years was detected. The 5.9-year periodicity continues to exhibit a degree of power in narrow time intervals, particularly in relation to PDO, NINO3, ESPI and MEI over 2000-2020. Additionally, the NAO over 1951-1990 and SOI over 1990-2010 also demonstrate this periodicity. However, as previously stated, the wavelet spectra is still incapable of resolving a stable oscillation of ~5.9 years. It can be seen that the classical Fourier spectrum, which is restricted to frequency resolution, occasionally exhibits unreliable low-frequency signals, which may arise from the superposition of near-periodic signals in different time spans. Hence, in order to acquire precise information pertaining to a long-period signal, it is more efficacious to scrutinize its instantaneous fluctuations.

[Figure]

**Figure 8. NOAA Climate Indices:** The Morlet wavelet (left) and Fourier (right) spectra of the climate monthly index data series since 1951 to present. From top to bottom: Pacific Decadal Oscillation (PDO), Arctic Oscillation (AO), North Atlantic Oscillation (NAO), Southern Oscillation Index (SOI), Atlantic Multi-decadal Oscillation (AMO), Eastern Tropical Pacific SST (NINO3), Antarctic Oscillation (AAO), ENSO Precipitation Index (ESPI), and Multivariate ENSO Index (MEI).

Fig. 9 provides evident indication that there are no discernible peaks present at ~5.9 years in either the Fourier or Morlet wavelet spectra of the GMST anomaly data series. The horizontal white lines in the wavelet spectra show the resolved 18.6-year lunar tide signals (Fig. 9c-f), which are also identified in the Fourier spectra (Fig. 9b). The Fourier spectra also reveal the presence of notable peaks at the periods of ~10 years, which are in proximity to the 11-year oscillation albeit indiscernible in Fig. 9c-f. It is noteworthy that there exist peaks of ~6.3 years in the Fourier spectra, which align with the findings of Pfeffer et al. (2023) as depicted in their Fig. 8. However, these peaks have been confirmed to be fake signals in the wavelet spectra.

[Figure]

**Figure 9**. **Global mean sea temperature (GMST):** The Fourier amplitude spectra (b) of the detrended annual GMST anomaly data series (a). Data source: GMST_NCDC (from NOAA/NCDC), GMST_GISS (from NASA/GISS), GMST_JMA (from Japan Meteorological Agency), GMST_HadCRUT5 (from Met Office Hadley Centre observations datasets).

Fig. 10 depicts the Fourier and Morlet wavelet spectra of the GMSL data series obtained from various sources, including CSIRO, JPL, AVISO, NOAA, Colorado, and NASA. In the analyzed data sets, specifically in the monthly reconstructed data from CSIRO and JPL and 10-day sampling data from

AVISO with seasonal signals removed, no discernible peaks were observed around the 5.9 years (Fig.

10c-e). In the time series from NOAA, Colorado, and NASA spanning from 1993 to present, with seasonal signals retained, the very weak peaks around 5.9 years are detected in the Fourier spectra (Fig.

10f-h). These peaks are significantly lower than the 95% CLs. Furthermore, we do not find any statistically significant signals exceeding the 95% CL around 6-7 years, especially 6.3 years, which were exhibited by Moreira et al. (2019) and Pfeffer et al. (2023) in the power spectra density periodogram.

[Figure]

**Figure 10**. **Global mean sea levels (GMSL):** The Fourier (b-h) and Morlet wavelet (i-n) spectra of the GMSL data series (a). (c, i) Monthly reconstructed data from CSIRO (Common-wealth Scientific and Industrial Research Organization), 1880-2014; (d, j) Monthly reconstructed data from JPL (Jet Propulsion Laboratory), 1950-2009; (e, k) 10-day interval data from AVISO of CNSE (Centre National d'Etudes Spatiales), 1993-2020; (f, l) 10-day interval data from NOAA Climate.gov, 1993-2020; (g, m) Monthly data from Sea Level Research Group of University of Colorado, 1993-2023; (h, n) 10-day interval data of TPJAOS v5.1 (Integrated Multi-Mission Ocean Altimeter Data) from the NASA Sea Level Change program, 1993-2023.

**(4) Verification using the oblateness $\Delta J_2$**

The terrestrial water storage variations can result in Earth's mass redistribution, potentially causing the change of the Earth's shape (via private communication with Benjamin F. Chao). As per this perspective, the $J_2$ variations ($\Delta J_2$), which serve as indicators of alterations in the oblateness of the Earth, have the potential to reflect the global hydrological changes. Therefore, we use the Fourier and wavelet spectra to analyze the $\Delta J_2$ time series spanning from 1975 to 2023 (https://filedrop.csr.utexas.edu/pub/slr/degree_2/). Fig. 11 demonstrates notable signals on interannual-to-decadal timescales., i.e., the 18.6-year lunar tide and 11-year variation, which highly coincide with the results of Chao et al. (2020). These signals are also found in the spectra analysis of the ERA5-Land global precipitation model (see Fig. 6). Additionally, the Fourier spectrum reveals the presence of a weak and spurious signal of a ~5 years period, which is consistent with the findings in Fig. 6. This exemplary correlation serves to illustrate that the oblateness $\Delta J_2$ has the capacity to depict certain overarching hydrological information on a global scale. The absence of a SYO signal in $\Delta J_2$ suggests a lower probability of the hydrological effects being the source of the SYO signal.

[Figure]

**Figure 11. ΔJ₂:** The Morlet wavelet (b) and Fourier (c) spectra of the $\Delta J_2(t)$ data series (a) in 1975-2023. The oblateness $J_2$ is the Earth's lowest-degree gravitational component that measures the (normalized) difference between the polar and equatorial moments of inertia. The $J_2$ time series is concatenated from satellite laser ranging data (Cheng et al., 2004). The 18.6-year lunar tide, 11-year and 33-year fluctuations are prominent, but no appreciable presence of SYO is detected, ruling out a degree-2 order-0 mass change (a redistribution in the net meridional sense) for causing the SYO in hydrological effects (Ding et al., 2018, EPSL, Supplementary Materials).

Besides, the authors still employ the AR-z spectrum. This method has been used in many papers now, but the code is still not made publicly available. If it is so much better than the FFT, why you do not share it? This AR-z spectrum always displays additional peaks with respect to the FFT. How do you explain the additional peaks that are visible on Fig. 3 but not visible in FFT?

Response: Ding et al. (2018, JGR) have provided the test code of the AR-z spectrum in their Supporting Information (https://agupubs.onlinelibrary.wiley.com/doi/10.1029/ 2018JB015890). A Matlab code for the AR-z spectrum, which includes a function code 'arz_spec.m' and a test code 'test.m' accompanied by a detailed description, was recently made available to the public on ResearchGate (https://www.researchgate.net/publication/370231571_The_simply_Matlab_code_for_the_ARz_spectrum). The methodology was executed and assessed by Hsu et al. (2021, JoG, https://doi.org/10.1007/s00190-021-01503-x). The present study employed the stabilized AR-z spectrum technique, which incorporated a Monte Carlo noise-assisted bootstrap scheme to yield more robust spectral estimates for a single record. It should be noted that the shared codes and the one verified by Hsu et al. (2021) are consistent as the core code to implement the AR-z method, without considering the noise-adding process; whereas this noise-adding process for the stabilized AR-z spectrum can be easily implemented (refer to Ding et al., 2018, JGR for details).

The AR-z spectrum exhibits several peaks with high SNRs, which are notably distinct from those observed in the FFT spectrum. However, it is imperative to underscore that the power of the peaks in the AR-z spectrum directly correlates with their stability rather than their actual amplitude. Please refer to Ding et al. (2018, JGR)'s supporting information for the conclusion. In other words, the spectral peak in the AR-z spectrum may appear strong even if the signal is weak, provided that it is relatively stable. As seen in Fig. 3 of the manuscript, the atmospheric/oceanic/hydrological (AOH) signals in the 3-5 years frequency band are representative instances. The AR-z spectra of the CA, MO, and ST stations exhibit insignificant spectral peaks, suggesting that the AOH signals are quite erratic during the analysed time spans. This is also discernible in the wavelet spectra, as illustrated in Fig. 4 of the manuscript.

P. 11, section 4.2: the argument of the ratio $\delta/h$ is not sufficient to propose an internal origin for the 6-yr oscillation. This ratio for surface loading is also very different from the tidal one (see for instance de Linage et al. 2007, doi: 10.1111/j.1365-246X.2007.03613.x and de Linage et al. 2009, doi: 10.1111/j.1365-246X.2007.03613.x, who have estimated this ratio for various loading and have shown some variability). Local hydrology would also affect this ratio, particularly at underground stations like Moxa, Membach, Strasbourg (e.g. Rosat et al. 2020, https://doi.org/10.1007/1345_2020_117). This argument is hence not sufficient to justify your interpretation of the 6-yr oscillation as the signature of an internal process.

**Response**: We first appreciate your valuable comments and providing us with significant references.

The degree-2 tidal Love numbers were exclusively taken into account to derive the ratio of $\delta/h$, which was determined to be approximately 1.9, while the surface loading ones were disregarded. This is an important oversight in the manuscript.

The actions of surface gravity variations resulting from a surface load comprise of the load's direct attraction and elastic deformation, and the latter also encompasses mass redistribution and free-air effect (de Linage et al., 2007). According to de Linage et al. (2009), the average ratio for hydrological loading (which includes soil moisture and snow) over the continents is -0.87 µGal mm$^{-1}$, but this ratio tends to increase as the size of river basins decreases; The atmospheric loading, assuming an inverted- barometer response of the ocean, exhibits larger values for high latitudes, with a positive ratio of 0.49

µGal mm$^{-1}$ (because the atmospheric masses are located above the measurement point); In the case of ocean tidal loading, the mean ratio for diurnal tidal waves over the continents is -0.26 µGal mm$^{-1}$. The relationship between vertical deformation and surface gravity, as expressed by $\Delta g = -\frac{2g}{R}\frac{\delta}{h}\Delta V$, allows for the approximate determinations of the ratios $\delta/h$ associated with the hydrological, atmospheric, and ocean tidal loadings. Specifically, the ratios corresponding to these loadings are approximately 2.84, 1.60, and 0.85, respectively. In contrast with our calculated values of 2.0 to 4.1, the atmospheric and oceanic tidal loadings as the external sources of the SYO can be excluded, whereas the hydrological loading's contribution to the SYO still needs intensive discussions. However, as demonstrated in the above responses, we have confirmed that the hydrological loading has a negligible impact on the surface gravity variation linked to the SYO.

Hence, it can be concluded that we can rule out that the SYO originates from external sources, but attribute it to some internal dynamical processes, such as the MAC wave we suggested; the 6-year related gravity changes, which may include core motions and some unknown 6-year changes due to strong coupling interactions between the mantle and core. Namely, the 6-year related surface gravity changes may be the result of a superposition of multiple internal motions in the Earth's coupling layers.

In fact, we have stated this opinion in the 2nd paragraph (lines 265-267) in Section 5, i.e., "we assume that the Earth's core processes may also be coupled to the motions of other regions in the Earth, and thus form secondary effects; And the magnitudes of which are not necessarily smaller than the direct effects." Except excluding the external loads, our conclusions are similar to the conclusions of Cazenave et al. (2023) and Pfeffer et al. (2023), who suggested the SYO affects the Earth system as a whole. These new discussions and references will be added to the revised manuscript.

Lines 270-272: the statement here is wrong. In Gillet et al. (2020) they used pressure Love numbers exactly as in Greff-Lefftz et al. (2004). You can check the values for the Love numbers h in their respective Table 1 and see that they are the same. The mistakes are in Fang et al. (1996) who have considered the pressure flow as a surface load but they have ignored the deformation of the equipotential surfaces in the core. They only considered the deformation of the mantle, while in Greff-Lefftz et al. (2004) and in Gillet et al. (2020), they both considered the deformation of the mantle and of the equipotential surfaces in the core. The Love numbers and surface deformation estimates by Gillet et al. (2020) are hence correct.

**Response**: We have carefully re-read the articles by Gillet et al. (2020), Greff-Lefftz et al. (2004) and Fang et al. (1996). Indeed, as you say, we have made a nonnegligible mistake. We will remove these wrong discussions in the revised manuscript.

Technical corrections

Lines 51, 59, 64 etc… satellite laser ranging should be abbreviated as SLR not SRL

**Response**: Thank you for your correction for our clerical error, and we will revise the manuscript.

Line 73: the GGP project does not exist anymore, it has been replaced by the IGETS.

**Response**: We appreciate your corrections regarding the incorrect description in our manuscript, and we will proceed to make the necessary revisions accordingly.

Lines 86-87: you say that you used level-2 products that mean that major disturbances have already been corrected from the data. Else, please precise what you are referring to as "h2" corrections since official IGETS products are called Level 1, Level 2 and Level 3 products.

**Response**: We must explain that the Code "h2" dataset, which is collected in the Level 2 data products, includes hourly gravity and pressure data corrected for instrumental perturbations and ready for tidal analysis (see Boy et al., 2020), was adopted for the used records (see Supplement Table S1 for the detailed information). We will modify the manuscript to avoid unnecessary misunderstanding.

Line 154: in the processing of data, a tidal analysis was performed with ETERNA software to remove tides. So why is there still the 18.6-yr tide? You did not include it in the groups of waves to be analyzed? Why?

**Response**: We must declare that the long-period tide constituents, ranging from SA to MQM tides, have been considered in our tidal analysis. However, as analyzed by many scholars, the harmonic analysis results of long-period tides are not very ideal, i.e., exhibiting significant amplitude and phase errors. This is attributed to that the time length is not enough for extracting precise information of long-period tide through iterative least-squares. Therefore, we employed a simple approach by substituting the long-period tide constituents with the 'long' wave, which operates within the frequency range of 0.004709-0.501369 cpd, and exhibits an amplitude and phase of 1.15000 and 0, respectively, as recommended by Tsoft. This method includes the 18.6-year tide and also leaves the intradecadal fluctuation unaffected.

Line 160: some spurious or unexplained peaks are visible in the AR-z spectrum (between annual and QBO, and at 2.6-yr). Why you do not discuss them? Are they artefacts of the AR-z spectrum? How confident are you on the AR-z spectral peaks? You should provide some confidence levels with this method, since many spurious peaks seem to appear…

**Response**: The present study focuses on the analysis of interannual-to-decadal signals, as detailed in Section 3. The signals within the frequency band of 1-2 years were not discussed, as they were considered as background noise. The peaks in this frequency band were amplified to an observable level because of the usage of an analytical continuation distinct from that employed in the interannual-to-decadal band. Certain peaks may possess practical significance or could potentially arise from background noise. Evidently, our unreasonable handling led to your misunderstanding. In the revised manuscript, we will conduct a more meticulous analytic continuation of this frequency band to align with the Fourier spectrum. Alternatively, we will employ a more straightforward method of low-pass filtering during data preprocessing to eliminate the frequency band above 2 cpy and minimize the influences of unidentified signals. Besides, it should be noted that during the implementation of nonlinear fitting for the recovery of the ~5.9-year oscillation, the frequency band of 1-2 years was not considered due to its negligible impact on the retrieval process. Finally, we appreciate your reminder, and the confidence levels for the AR-z spectra will be added in the revised manuscript.

**References**

Boy, J.-P., Barriot, J. P., Förste, C., Voigt, C., and Wziontek, H.: Achievements of the first 4 years of the International Geodynamics and Earth Tide Service (IGETS) 2015-2019. In: Freymueller, J. T., Sánchez, L. (eds) Beyond 100: The Next Century in Geodesy. International Association of Geodesy Symposia, vol 152. Springer, Cham. https://doi.org/10.1007/1345_2020_94, 2020.

Cazenave, A., Pfeffer, J., Mandea, M., and Dehant, V.: ESD Ideas: A 6-year oscillation in the whole Earth system?, EGUsphere [preprint], https://doi.org/10.5194/egusphere-2023-312, 2023.

Chao, B. F. and Yu, Y.: Variation of the equatorial moments of inertia associated with a 6-year westward rotary motion in the Earth, Earth Planet. Sci. Lett., 542, 116316, https://doi.org/10.1016/j.epsl.2020.116316, 2020.

Chao, B. F., Yu, Y., and Chung, C. H.: Variation of Earth's oblateness $J_2$ on interannual-to-decadal timescales, J. Geophys. Res. Solid Earth, 12, e2020JB019421, https://doi.org/10.1029/2020JB019421, 2020.

Cheng, M. K. and Tapley, B. D.: Variations in the Earth's oblateness during the past 28 years, J. Geophys. Res., 109, B09402, https://doi.org/10.1029/2004JB003028, 2004.

Ding, H.: Attenuation and excitation of the ~6-year oscillation in the length-of-day variation, Earth Planet. Sci. Lett., 507, 131-139, https://doi.org/10.1016/j.epsl.2018.12.003, 2019.

Ding, H., An, Y., and Shen, W.: New evidence for the fluctuation characteristics of intradecadal periodic signals in length-of-day variation, J. Geophys. Res. Solid Earth, 126, e2020JB020990, https://doi.org/10.1029/2020JB020990, 2021.

Ding, H. and Chao, B. F.: A 6-year westward rotary motion in the Earth: Detection and possible MICG coupling mechanism, Earth Planet. Sci. Lett., 295, 50-55, https://doi.org/10.1016/j.epsl.2018.05.009, 2018.

Ding, H. and Chao, B. F.: Application of stabilized AR‐z spectrum in harmonic analysis for geophysics, J. Geophys. Res. Solid Earth, 123, 8249-8259, https://doi.org/10.1029/2018JB015890, 2018.

Duan, P. S., and Huang, C. L.: Intradecadal variations in length of day and their correspondence with geomagnetic jerks, Nat. Commun., 11(1), 2273, https://doi.org/10.1038/s41467-020-16109-8, 2020.

Gillet, N., Jault, D., Canet, E., and Fournier, A.: Fast torsional waves and strong magnetic field within the Earth's core, Nature, 465(7294), 74, https://doi.org/10.1038/nature09010, 2010.

Hsu, C. C., Duan, P. S., Xu, X. Q., Zhou, Y. H., and Huang, C. L.: On the ~7 year periodic signal in length of day from a frequency domain stepwise regression method, J. Geod., 95, 1-15, https://doi.org/10.1007/s00190-021-01503-x, 2021.

Liao, D. C. and Greiner-Mai, H.: A new ΔLOD series in monthly intervals (1892.0-1997.0) and its comparison with other geophysical results, J. Geod., 73, 466-477, https://doi.org/10.1007/PL00004002, 1999.

Moreira, L., Cazenave, A., and Palanisamy, H.: Influence of interannual variability in estimating the rate and acceleration of present-day global mean sea level, Global Planet. Change, 199, 103450. https://doi.org/10.1016/j.gloplacha.2021.103450, 2021.

Pfeffer, J., Cazenave, A., Blazquez, A., Decharme, B., Munier, S., and Barnoud, A.: Detection of slow changes in terrestrial water storage with GRACE and GRACE-FO satellite gravity missions, EGUsphere [preprint], https://doi.org/10.5194/egusphere-2022-1032, 2022.

Pfeffer J., Cazenave A., Moreira L., Rosat S., Mandea M. and Dehant V.: A 6-year cycle in the Earth system, SSRN Electronic Journal, https://doi.org//10.2139/ssrn.4388237, 2023.

de Linage, C., Hinderer, J., and Rogister, Y.: A search for the ratio between gravity variation and vertical displacement due to a surface load, Geophys. J. Inter., 171(3), 986-994, https://doi.org//10.1111/j.1365-246x.2007.03613.x, 2007.

de Linage, C., Hinderer, J., and Boy, J. P.: Variability of the gravity-to-height ratio due to surface loads, Pure Appl. Geophys, 166, 1217-1245, https://doi.org/10.1007/s00024-004-0506-0, 2009.

Zotov, L., Bizouard, C., Sidorenkov, N., Ustinov, A., and Ershova, T.: Multidecadal and 6-year variations of LOD, J. Phys.: Conf. Ser., 1705, 012002, https://doi.org//10.1088/1742-6596/1705/1/012002, 2020.

---

## Author Comment (AC3)

We appreciate your comments once more. Our responses to your questions are as follows:

(1) Previous evidences from different observation data (e.g., ΔLOD, geomagnetic field, GNSS, and polar motion) have indicated that the SYO in ΔLOD is an almost stable oscillation with attenuation larger than 180. This suggests that the SYO persists throughout the entire LOD observation period from 1962 to the present. Thus one should use the observations for an extended duration, preferably in alignment with the ΔLOD time, to verify the existence of a steady ~5.9-year fluctuation before linking it with the SYO in ΔLOD. Moreover, an unstable oscillation signal should not be considered cognate with the SYO in ΔLOD, even though there exists the oscillation energy around 5.9 years in a relatively short time, as observed in Fig. 3 (some stations seem to have signals of ~4-8 years in the SG time windows). In other word, it is possible to say that the hydrology makes a transient contribution to ΔLOD. This is obviously not true. In addition, if the SYO had been generated by the hydrological effects, it would have been evident in the hydrologic observation data over an extended time and in different regions. However, this assertion is contradicted by the findings of Pfeffer et al. (2022) and Pfreffer & Cazennave et al. (2023). From the wavelet spectra analysis, we think it may be a result of noise interference, or energy leakage from nearby signals.

Additionally, you consider it reasonable that the inaccuracy of the current hydrological models is the reason why the possible SYO peaks in the spectra of the hydrological loading data do not precisely exactly align with the 5.9 years period. However, it must be founded on a precondition that the SYO is actually present in the hydrological models. Actually, this precondition has not yet been confirmed precisely because of the inaccuracy of the hydrological model. Your discussions are obviously illogical.

(2) Regarding the verifications using global gridded precipitation data, climate indices, GMST, and GMSL, you both posed questions similar to that in validation using hydrological models. Specifically, all of these validations should be conducted using smaller time windows as for SGs. It is imperative to reiterate that it is necessary to use observations for extended durations, preferably in alignment with the ΔLOD time, to ascertain the presence of a stable ~5.9-year fluctuation before establishing a correlation between it with the SYO in LOD.

(3) Observing Fig. 4(c) in Rosat & Gillet (2023), an extremely weak 5.9-year signal with an amplitude of ~0.012 ms is present in the wavelet transform coefficient spectrum of the HAM data. Nevertheless, it can be observed that the amplitude is nearly congruent with the background noise level of ΔLOD.

As depicted in Fig. 1, the background noise level of ΔLOD after removing the AAM and OAM effects is ~0.01 ms. It also can be seen in Fig. 1d of Ding (2019, EPSL) and Fig. 1b of Ding et al. (2021, JGR).

That is to say, despite the presence of a ~5.9-year signal in the hydrology-excited LOD, the associated peak is likely to be buried in the background noise (the SNR is very low). Statistically, the signal is unreliable.

[Figure]

**Figure 1**. The Fourier amplitude spectra of the original ΔLOD in 1962-2021 (in gray), and the residual (in blue) obtained from the original ΔLOD after removing the AAM and OAM effects.

[Figure]

**Figure 2.** The Fourier amplitude spectra and wavelet spectra of two synthetic white noise time series.

We also carried out a series of synthetic tests. We simulated the random white noises with the mean amplitudes equivalent to the background noise level of ΔLOD in Fig. 1, and subsequently conducted the Fourier and wavelet spectrum analysis on them. Fig. 2 shows two test results for the synthetic white noises. Obviously, the wavelet spectra display clear ~6-year oscillation signals that are definitely fake, while the Fourier amplitude spectra do not manifest any noteworthy signal at all.

Thus, purely from the point of signal analysis, it is reasonable to conclude that the hydrology-excited

LOD time series does not contain the SYO signal. To avoid unnecessary misunderstandings, it may be more appropriate to rephrase the statement as follows: The HAM effect made very small contributions to the intradecadal period band in the ΔLOD.

(4) We disagree your claim about the influences of the terrestrial water storage variations on $\Delta J_2$. The change in the zonal harmonic coefficients $\Delta J_2$, of the Earth's gravitational field due to the surface- water-induced mass redistribution can be calculated by (Chao et al. 1988):

$$\Delta J_l(t) = -\frac{R^2 \rho}{M}(1 + k_l') \int [h(\theta, \lambda, t) - \bar{h}(\theta, \lambda)]P_l(\cos\theta)d\Omega \qquad (1)$$

where $M$ is the mass of the Earth, and $P_l$ is the Legendre polynomial of degree $l$; $k_l'$ are the load Love numbers; $\rho = 1$ g m$^{-3}$, $h$ is the equivalent depth of liquid water with average $\bar{h}$. The change in ΔLOD

due to surface mass redistribution is directly proportional to $\Delta J_2$, according to

$$\Delta\text{LOD} = \text{LOD}[2MR^2\Delta J_2(t)]/(3C) \qquad (2)$$

where $C$ is the moments of inertia about the Earth's polar.

It can be affirmed that the terrestrial water storage (TWS) variations can result in the C20 variations, which are directly reflected in ΔLOD. Besides, it is worth noting that Chao and Eanes (1995)

demonstrated global gravitational changes due to atmospheric mass redistribution, which can also manifest in the C20 variations. Therefore, if an SYO in the TWS variations is observed, it must appear in $\Delta J_2$, and further will be reflected in ΔLOD. However, in our last reply, it has been determined that there is no observable SYO signal present within $\Delta J_2$. Given the stable SYO signal present in

ΔLOD, the only explanation is that the SYO signal in ΔLOD does not have any correlation with the

TWS variations.

In addition, we have carefully read the relevant results in Meyssignac et al. (2013). You claimed that "The hydrological content at interannual time-scales is mostly related to degree-2 order-2 geographical pattern". This is definitely incorrect. Meyssignac et al. (2013) have clearly indicated that "*Variations in land water storage (hereafter LWS) also play a role in S2,2 variations (see Figure 1c). But both hydrological models agree to show that the LWS contribution to S2,2 interannual variations is rather small*", "*As a result, C2,2 variations estimated from SLR tracking data and the combination of ocean mass and LWS contributions are quite different. Nevertheless, we note that both estimations agree on showing small C2,2 variations*", and "*But unlike the ocean mass contribution, they show an important role of LWS on C2,0 variations over the whole record*." They showed that the hydrological content at interannual time scales is mostly related to degree-2 order-0 geographical pattern. This is the exact opposite of what you said.

Besides, you claimed that "Only a certain distribution of terrestrial water storage variations would result in $\Delta J_2$." It has no basis at all. According to Chao et al. (2020), "*$J_2$ is the (normalized) zonal quadrupole of the Earth's density.*" "*The zonal [degree-2, order-0] component of any mass redistribution will contribute to the time variation $\Delta J_2$. Besides the seasonal water cycle in the surface geophysical fluids (atmosphere + hydrosphere + cryosphere), a host of geophysical processes cause mass redistribution on/in the Earth ranging from tides to atmosphere-ocean circulations, to denudation of glaciers/ice sheets and sea level rise, and to internal phenomena like earthquakes, glacial isostatic adjustment (GIA), and core flows.*"

(5) The revised manuscript will present the results of implementing AR-z into some better hydrological models like ERA5_land.

(6) As demonstrated in our last reply, both current global hydrological models and global precipitation data do not indicate the presence of a stable and consecutive ~5.9-year oscillation. To verify the local contribution of hydrological mass changes, we can only use the in-suit groundwater level data from a few SG stations to to illustrate it. Fig. 3 shows the Fourier amplitude spectra of six groundwater level records, indicating no ~5.9-year signal exists. To get the accurate local contribution of hydrological mass changes, more extensive global in-suit hydrological observations (including precipitation and soil moisture) should be joint for comprehensive verification in the future.

[Figure]

**Figure 3**. The Fourier amplitude spectra of six groundwater level records.

(7) Indeed, as you said, the pre-processing process, especially the repair of gaps and steps, is the main challenge to detect long-period signals using SG data. Manual corrections using Tsoft will lead to significant differences of the Fourier spectra results due to operational differences. So it is not surprising that you observed the 4-yr, 5-yr, or 7-yr periodic signals in different SG residual series. To address this unavoidable problem, we have conducted a very careful pre-processing work (see Fig.1 in the manuscript) and analyzed the errors of step corrections (see our Supplement).

You found the SYO clearly for Strasbourg and Metsahovi with a nice anti-correlation with the hydrological loading. In fact, thus good negative correlations were also observed by us. However, it is not sufficient to prove that there is an anti-phase SYO in hydrology corresponding to that in the SG residuals. Because we have verified that there are irregular fluctuations at ~4-8 years in the hydrological model data, which makes the time series appear to be inversely approximate to the long-period fluctuations in the SG residuals.

In addition, we can only obtain the SG data with lengths about 3-4 cycles of the SYO period, which are enough to describe the SYO information. Any unreasonable step correction (very different from the actual situation) will cause a significant deviation of the SYO spectral peak in the Fourier spectrum.

**References**

Chao, B. F. and O'Connor, W. P.: Global surface-water-induced seasonal variations in the Earth's
rotation and gravitational field, Geophysical Journal, 94, 263-270, 1988.

Chao, B. F. and Eanes, R.: Global gravitational changes due to atmospheric mass redistribution as
observed by the Lageos nodal residual, Geophys. J. Int., 122,755-764, 1995.

Chao, B. F., Yu, Y., & Chung, C. H.: Variation of Earth's oblateness J2 on interannual-to-decadal
timescales. J. Geophys. Res.: Solid Earth, 125, e2020JB019421, https://doi.org/10.1029/
2020JB019421, 2020.

Ding, H.: Attenuation and excitation of the ~6-year oscillation in the length-of-day variation, Earth
363 Planet. Sci. Lett., 507, 131-139, https://doi.org/10.1016/j.epsl.2018.12.003, 2019.

Ding, H., An, Y., and Shen, W.: New evidence for the fluctuation characteristics of intradecadal 365
periodic signals in length-of-day variation, J. Geophys. Res. Solid Earth, 126, e2020JB020990,
366 https://doi.org/10.1029/2020JB020990, 2021

Meyssignac, B., Lemoine, J. M., Cheng, M., Cazenave, A., Gégout, P. and Maisongrande, P.:
Interannual variations in degree-2 Earth's gravity coefficients C2,0, C2,2, and S2,2 reveal large-
scale mass transfers of climatic origin, Geophys. Res. Lett., 40, 4060-4065,
https://doi.org/10.1002/grl.50772, 2013.

Rosat, S. , and Gillet, N.:  Intradecadal variations in length of day: Coherence with models of the
Earth's core dynamics, Phys. Earth Planet. Inter., 41, 107053, 2023.

---

## Author Comment (AC4)

**Response to Referee #2**

**1 General comments**

The paper raises the interesting question if a 5.9 year oscillation (SYO) observed in other geodetic time series, can also be detected in SG data. I think this is a relevant question and it might help to interpret the origin of such an oscillation.

The authors come to the conclusion, that they can detect SOY signals in the data and estimate their amplitudes and phases.

However, the preprocessing of the data, especially the correction of instrumental disturbances, is very critical, when analysing very long time series of SG data. Especially, for the step correction I have doubts if the methods used in this paper are completely valid. Some of the methods also need more explanation. Further, the accuracy of these corrections might be overestimated by the authors, which highly affects the detection and the interpretation of the 5.9 year oscillation with the small amplitude of 5 nm/s$^2$ to 9 nm/s$^2$.

Further, as highly discussed between the authors and Reviewer 1, the origin of such an oscillation might still be debatable. As I see my expertise rather in the processing of SG data, I don't want to add many comments to this discussion.

**Response**: We appreciate your valuable comments and suggestions that may lead to significant improvements of the manuscript. As previously discussed with Reviewer 1, in this study, there are two questions that really deserve in-depth investigation: (1) Dose the SYO originate from an internal or external source? (2) How to conduct the reasonable SG data pre-processing, especially for repairs of data steps and gaps, which will seriously affect the retrieval of annual-to-decadal fluctuations? Regarding Question (1), we had detailed discussions with Reviewer 1, and we were greatly inspired. Regarding Question (2), we had preliminary discussions with Reviewer 1, and we also emphasized that we had taken great care in the pre-processing work to ensure the correctness of the SYO signal extraction. Here, we will provide a detailed response to your comments related to Question (2).

**2 Specific comments**

**2.1 Data source**

I would like to comment on the confusion about the "h2" database of IGETS.

For most of the stations in the IGETS database three different kinds of Level 2 data exist. They are specified by the code in the ending of their filename: 22.gpp, 32.gpp and h2.gpp. The meaning of the code "22" and "h2" is documented by Voigt et al. (2016). The procedure to produce the "32"-data is explained by Boy et al. (2023), however the code "32" is only mentioned by Boy (2022).

The "22"-files should contain data with a sampling interval of 1 min, where gaps and disturbances are filled with synthetic data and offsets are adjusted, but it seems (see below) that this is at least not true for big gaps and big offsets. The "h2" files contain hourly sampled data, but Voigt et al. (2016) do not specify which kind of preprocessing was performed. The comparison below indicates that it is the same as for the "22" files.

The preprocessing for "22" and "h2" data is done at UFP (University of French Polynesia).

The "32" files are produced by EOST (Ecole et Observatoire des Sciences de la Terre). They contain data with a sampling interval of 1 min, where gaps are filled and offsets are adjusted.

To clarify, I provide a comparison of the calibrated Level 1 data and all kinds of Level 2 data for the year of 2006 and the gravimeter SG026 at Strasbourg. It is shown in Figure 1. It can be seen that the data spikes in November were removed in all versions of level 2 data. All gaps were filled and steps were adjusted in the "32"-data, while this is not the case for "22" and "h2" data. From visual inspection the "h2" seems to be a downsampled version of "22" data.

As you use your own method to adjust steps, I think it is reasonable that you use a data set, where steps were not adjusted before, which might be the case for "h2" data although it is stated differently by Voigt et al. (2016). In line 86 of your manuscript you mention that you also removed spikes. Does this mean that there were remaining spikes in the "h2" data that you removed on your own? Please clarify on this. If you anyway correct spikes and steps on your own, I think the safest option would be to use the level 1 data. Then you would know that no preprocessing was done and an uncertainty about the preprocessing done by IGETS cannot influence your results.

Please clarify in your manuscript which datasets from IGETS you use (level and code) and which preprocessing steps where done by IGETS.

**Response:** We fully endorse your description of the data set published by IGETS. Indeed, there are some differences between the actual published SG data and declaration by Voigt et al. (2016), especially for the Level 2 products. Voigt et al. (2016) and Boy et al. (2019) claimed that the Level 2 products are the data corrected for gaps, spikes, and steps. Actually, it is not true for big gaps and big offsets (you mentioned above too). That is where we get confused. We used the "h2" data, focusing on the corrections of big gaps and big steps, and with no correction of spikes. We are very sorry for our mistake, that caused your misunderstanding. The corrections of the spikes from the raw data to "h2" has been almost completely conducted.

Indeed, as you said, the most reasonable pre-processing way is to start with the "00" data, and the integrity of the data can be guaranteed. We initially did consider using the "00" data; however, the implementation proved to be exceedingly challenging. The minutely "00" data are riddled with numerous spikes, gaps, and steps, which are from the instrument problems and environmental effects. The method of IGETS to remove part of them by filling with synthetic data should be a relatively simple and reasonable way at present. Therefore, we ended up using the "h2" data, with the elimination of most of gaps, spikes, and steps in the IGETS processes with synthetic tides. However, some big steps were retained. That is what we focused on analyzing. We were very careful in repairing the steps, and we also did the error analysis to ensure the reliability.

In the revised manuscript, we will add more description related to the used SG dataset, including data information and preprocessing steps where done by IGETS, according to your suggestion.

**2.2 Step correction**

I have some questions and comments on your step detection and step size determination process.

First, I consider it a new and interesting idea to estimate the step sizes by fitting the data to the polar motion times series. I have not seen this approach before. However, I have some doubts if it is completely valid.

…

Therefore, I would propose to estimate the size of the steps, and all amplitudes and phases of the
periodic signals at the same time. Otherwise you have to justify, why your method is still valid, or, if
it is not completely valid, how big the errors are introduced by using it anyway.

Further, please also explain the method in more detail in your paper. As you write in the end of the
paper, the best would be to also include absolute gravity measurements in the step size and drift
estimation.

For the smaller steps you use a second method to find and remove them, which is explained in the
supplementary material. From there it is very clear what you did to get to the green curve in Figure
S2(c). However, please explain how you determine the step times and sizes from the green curve. This
is not clear.

Especially, for the bigger steps in SG data a problem in estimating their size is that they often occur
together with data gaps and that sometimes a running-in behaviour of the data occurs after the step.
How is the accuracy of your step removing procedures influenced by this problem?

Beside the question if and how hydrology needs to be removed, the step correction is the biggest
uncertainty in your data processing. Therefore, I appreciate that you analyse the influence of errors in
the step corrections on your results. However, how do you come to the conclusion, that continuous
downward steps cause the maximal amplitude deviation? I did a synthetic test similar to the one you
present in the supplementary material, but for a mixture of upward and downward steps: I simulate a
synthetic time series containing white noise with a standard deviation of 10 nm/s$^2$ (Figure 2(a)), six
steps of 4 nm/s$^2$, which are upward and downward (Figure 2(b)) and a SOY signal with a period of 5.9
years and an amplitude of 6 nm/s$^2$ (Figure 2(c)). To simplify the synthetic test, I only estimated the
amplitude of the SOY in the synthetic dataset with a linear least squares estimation and assumed the
phase to be known. The simulated time series, the input SYO signal and the estimated SYO signal are
shown in Figure 2(c). The estimated amplitude of SYO in the synthetic data is 8.1 nm/s$^2$, which means
a deviation of more than 2 nm/s$^2$ from the input value. It is possible that the deviation is even bigger
for other configurations. This test should just show that the deviation can be bigger than 1.8 nm/s$^2$.
Furthermore, your synthetic test is only meaningful, if you can be sure that errors in your step determination and step correction process are not bigger than 4 nm/s$^2$. How did you come to this estimate?

In conclusion, I think the accuracy of the step correction might be overestimated. This leads to an underestimation of the errors of the SOY signal's amplitudes and phases, which finally affects the interpretation of the SOY signal.

**Response:** Many thanks for the detailed mathematical formula explanation. In accordance with the equation (1) you have presented, it is indeed the most rigorous approach to consider all potential steps throughout the entirety of the data analysis period. This entails estimating the sizes of the steps, as well as the amplitudes and phases of the periodic signals concurrently. However, the current operational approach is impractical, because of the existence of numerous gaps within the time series, thereby dividing the data series into distinct segments. Hence, we first identified the suitable segments based on the presence of gaps, and then piecewise fit the PM+ΔLOD series to the SG residual data (which has been subjected to filtering and smoothing) one by one. In this process we used an iterative search method. Given the relatively short time span of approximately 20 years, it is reasonable to approximate the long-term fluctuation as a linear trend and incorporate it into the fitting procedure. Naturally, this methodology falls short of attaining the level of rigor you have described, bur this approach just serves as a preliminary correction, primarily targeting significant gaps that are readily discernible to the unaided eye. Considering the need for small step correction in the next step, it is possible that the current correction, although imperfect, can be adjusted in subsequent correction. Besides, it is worth mentioning that we also employed the TSAnalyzer software developed by Wu et al. (2017), which is used for preprocessing the GPS data. This software is commonly utilized for correcting significant steps in the initial stages. However, the outcomes obtained from this approach were not as satisfactory as our current method. The small steps were identified according to the average fluctuations before and after a suspected step with respect to the overall fluctuation during the analysis segment. The process will be elucidated further in the subsequent revision.

For the synthetic test, we found the reason for the difference between our synthesis result and yours.

It is mainly attributed to the different time points of the first step we two set. We chose the same step occurrence time as you, and got the same result as you. Hence, we redesign a random synthesis test, which makes the input white noise (with a noise level near or lower than those of the SG residuals)

random, the initial input SYO phase random, the input step occurrence time random in every half SYO cycle, and the input step rise or fall random in every half SYO cycle. Under these conditions, we conduct 1000000 random trials for the simulated data within 18, 21 and 24 years (corresponding to 6, 7, and 8 steps of 4 nm/s$^2$), and calculate the deviation values of the amplitude and phase between the input and retrieved SYO (see an example in Figure R1). The statistical result shows that: 1) The percentages of the deviations less than 0.3 μGal are more than 99.5% for all three data lengths; 2) The percentages of the deviations less than 0.2 μGal are about 95%, 96%, and 97% respectively for all three data lengths; 3) The phase deviations are less than 0.15π. The results also shows that the longer the data length is, the less interference the steps have on the retrieval of the SYO. Therefore, from a statistical point of view, we believe the residual offsets may affect the observed ~5.9 years signal for the amplitude less than 0.2 μGal.

We sincerely appreciate you repeating the synthesis test and alerting us to the test's flaw, and we will modify the synthesis test and its relevant explanation in the revised supplementary material and manuscript. Some clarification you proposed will be also added in the revised manuscript.

[Figure]

**Figure R1**. Synthetic test for the effect of small steps on the SYO signal retrieval. (a) shows the simulated noise series, (b) shows the input steps, and (c) shows the simulated SOY, the simulated time series and the recovered SOY.

**2.3 Spectra**

You emphasise the better frequency resolution of the AR-z spectra compared to Fourier spectra. I agree that the frequency resolution of the Fourier spectra of $1/T \approx 0.5$ cpy is not enough to resolve signals in the 3-5 year band, the 8.5-18.6 year band or the SYO. Is it possible to quantify the frequency resolution of the AR-z spectra. If yes, please do so. This would help to know the accuracy of your determined frequencies.

Did you use any taper to compute the Fourier spectra? If yes, please specify.

**Response:** When applying the AR-z spectra method, we typically densify our spectral spacing by a factor of 3 (or 5) over the Fourier elementary spacing. It requires the calculation of discrete Fourier transforms in the AR solution, which in turn can be efficiently done by FFT with zero-padding (see details in Ding & Chao, 2015, GJI). We used the Hanning taper to compute the Fourier spectrum. The relevant information will be added in the revised manuscript.

**2.4 Retrieval of the SYO signal**

How do you finally obtain the periods of the SOY ranging from 5.84 to 5.92 years? (Line 188-189). Do you read them from the Morlet wavelet spectra or from the AR-z spectra? In Figure 5 you give uncertainties for these periods. How did you obtain them? Please specify.

For the same reasons as disused in Section 2.2, I think the estimates for amplitudes and phases of the SOY signal would be more stable, if you estimate them together with the amplitudes and phases of all the other periodic signals.

**Response:** The SYO period value was obtained by the Lorentz fitting estimation of the Fourier spectrum peak of the SYO after removing other long-period signals by least-squares fitting. The uncertainty is the estimation error, which is related to the background noise level.

We agree with the method that you suggested, and we have even experimented with it. However, in order to determine the estimated uncertainty, we ultimately employed the approach of initial elimination followed by fitting. The validity of this approach was further validated through simulation experimments. Please see Materials and Methods 3 in our Supplement.

**2.5 18.6 year period**

In the conclusion you claim you have identified an 18.6 year period in the SG data. However, in the Section 'Spectral analysis' you state correctly that the data length is too short to identify this oscillation.

**Response:** We appreciate you pointing out our significant mistake. The accurate determination of the signals within the entire 8.5-18.6 band is indeed lacking. The statement made in this context is incorrect, and it will be modified in the revised manuscript.

**2.6 Origin of the SYO**

It is not clear to me, why you think the SYO should originate from the Earth core. Even if you can exclude external and loading sources, why do you think the SOY more likely originates from the core than for example from the mantle?

**Response:** We must first demonstrate that the SYO is a long-period fluctuation on a global scale. So far, the discussions about the origins of thus Earth's interannual-to-decadal fluctuations mainly focuses on the surface process or deep interior dynamics, and the evidence from the solid mantle is very little. A large number of surface load observation data can provide the interpretation of some interannual signals; Other long-period signals that cannot be explained by the surface loads (see Ding, 2019, EPSL, for examples) are mostly attributed to the dynamic processes of the Earth's core, which are often controversial because the structure of the Earth's core is not well understood.

In this study, our main objective was to provide evidence from surface gravity monitoring, which is different from previous evidence from satellite observations (e.g., SLR and GRACE), for the core origin interpretation of the SYO. Here we are trying to find a more reasonable interpretation for our SYO gravity observations. Our core idea is that the SYO signal in ΔLOD is driven by core processes. The 6-year related gravity changes, which may include core motions and some unknown 6-year changes due to strong coupling interactions between the mantle and core. Namely, the 6-year related surface gravity changes may be the result of a superposition of multiple internal motions in the Earth's coupling layers.

**3 Technical corrections**

Line 32-46: A non-expert to core dynamics would have big difficulties to understand the basic ideas of this paragraph. In my opinion this would be fine for a paragraph in the discussion part. For an introduction, however, I think the basic ideas should also be understood by non expert readers. So it would be helpful if you could rewrite this paragraph in a less technical way.

**Response:** Previous discussions about the origin of SYO have mainly focused on the dynamic processes of the Earth's core, which are currently controversial. In this paragraph, we mainly wanted to briefly introduce some of the current conjectures or explanations for the SYO excitation mechanism. In the process of revision, we will carefully revise this paragraph with reference to your suggestions for the convenience of readers.

Line 49: (PM, Ding et al. 2019, 2021; Chen et al. 2019)→ (PM) (Ding et al. 2019, 2021; Chen et al. 2019). The same applies to line 51.

**Response:** Thank you very much for the technical correction. We have modified as follows: "In recent years, the fluctuation characteristics and excitations of the SYO have also been investigated using some continuous and long-span geophysical/geodetic observations, including the polar motion (PM) (Ding et al., 2019, 2021; Chen et al., 2019), GPS (Global Positioning System) displacements (Ding and Chao, 2018a; Watkins et al., 2018; Rosat et al., 2021), geomagnetic fields (Ding and Chao, 2018a), and gravity-field satellite laser ranging (SLR) (Chao and Yu, 2020; Rosat et al., 2021)."

Line 181: You write that you are fitting 8 harmonics: ~18.6/13.5, ~8.5, ~4.2, ~3.65, ~3.2, ~2.6, ~2 and ~1 year. What does '~' mean in this context? As I understand, you estimate the amplitudes and the frequencies by the fitting procedure, while the frequencies are fixed? If this is the case, which are the exact frequencies you are using? What do you mean by 18.6/13.5? Do you use both frequencies? Is it one frequency for each station?

**Response:** Here '~' refers to the meaning of 'about'. No special meaning! All these frequencies are from the estimated values using the AR-$z$ method in this study and previous empirical values (see our Section 3 in the manuscript). We were just keeping them one decimal place or two decimal places. This is true of all same symbols throughout the text. All these fixed frequencies associated with their cosine functions were used to fit each SG residual time series, to obtain their corresponding amplitudes and phases. Since the time is too short, we could not distinguish the peak with the period over 10 years, which may come from the overlap of adjacent spectral peaks. According previous empirical values, we chose two more reasonable values of 18.6 and 13.5 years to fit the peak regarded as a quasi-periodic oscillation. In practical fitting applications, we will prefer the value of 18.6 or 13.5 years or both, to achieve the best fitting of the SG time series from the perspective of the Fourier spectrum.

Line 259: When talking about pressure changes, are these air pressure changes or pressure changes inside the Earth? Please clarify.

**Response**: The pressure here refers to the non-hydrostatic pressure on the core-mantle boundary from the liquid outer core (Please see Dumberry, 2010, GJI). We will clarify it in the revised manuscript.

**References**

Ding, H.: Attenuation and excitation of the ∼6-year oscillation in the length-of-day variation, Earth Planet. Sci. Lett., 507, 131-139, https://doi.org/10.1016/j.epsl.2018.12.003, 2019.

Ding, H. and Chao, B. F.: Detecting harmonic signals in a noisy time series: The z-domain autoregressive (AR-z) spectrum, Geophys. J. Int., 201(3), 1287-1296, https://doi.org/10.1093/gji/ggv077, 2015.

Dumberry, M.: Gravity variations induced by core flows, Geophys. J. Int., 180(2), 635-650, https://doi.org/10.1111/j.1365-246X.2009.04437.x, 2010.

Wu, D., Yan, H. and Shen, Y.: TSAnalyzer, a GNSS time series analysis software, GPS Solut., 21, 1389-1394, https://doi.org/10.1007/s10291-017-0637-2, 2017.

---

## Author Comment (AC5)

**Response to RC3**

We express my sincere gratitude for your continued feedback. We agree with some of your points, and we will make appropriate modifications based on the information you offered. In addition, We still have the following two points to reply to you:

(1) The primary point of disputation in our discussions pertains to that whether the hydrological data contains the 5.9-year oscillation. The topic of selecting a time window remains a subject of ongoing discussion, and we will test various hydrological model data according to different time windows. Furthermore, subsequent to our discussions, a consensus has been reached that the existing hydrological models lack accuracy. The wavelet analysis findings presented in Reply on RC2 also corroborate the notion that certain long-period signals, such as those of 11 years and 18.6 years, exhibit instability and inaccuracy over extended time intervals. In addition, we support that the 5.9-year oscillation origins from the dynamics of the Earth's core, and further causes the mantle and crust connection effects. However, whether it also causes the fluid layers at the Earth's surface to produce the associated effects (especially the hydrological effect) remains to be discussed. We believe that to fully confirm whether the 5.9-year oscillation is contained in the hydrological processes, we need to carry out more detailed verification combined with global in-site hydrological observations. Therefore, we highly recommend taking on this work in the future.

(2) Regarding the validation of the AR-$z$ spectrum, we have shown in previous studies that the AR-$z$ spectrum is not suitable for direct application to a single time series, but for multiple time series in the form of the product spectrum (See Ding & Chao, 2015, GJI; 2015, JGR). Indeed, the AR-$z$ spectrum applied to a single time series yields the unstable result as you got in your tests. The AR-$z$ spectrum employs a preset $Q$ value for conducting analytical continuation, thereby enhancing certain spectral peaks that may potentially include unidentified noise. Therefore, based on the original AR-$z$ formulation, we have developed a stabilized AR-$z$ spectrum, taking advantage of a Monte Carlo noise-assisted bootstrap scheme, and demonstrate its effectiveness in obtaining more robust spectral estimates for a single record (Ding & Chao, 2018, JGR). This improved method can prevent the noise in a data record to alter the apparent estimates in the "empty" bins. Fig. R1 compares the Fourier spectrum, AR-$z$ spectrum, and stabilized AR-$z$ spectrum of a single synthetic time series. The synthetic signal and white noise are exactly the

same as the example you gave, and the $Q$ value used in the analytical continuation of AR-$z$ is 1e15 due to no attenuation for the input signal. From Fig. R1, the (stabilized) AR-$z$ spectrum exhibit the significant advantage in suppressing the background noise relative to the Fourier spectrum. Applying to a single time series, the AR-$z$ spectrum leads to some noise signals being mistakenly magnified because of its strong instability, whereas the stabilized AR-$z$ spectrum overcomes this drawback well.

It is also worth noting that the AR-$z$ method is more practical for the enhancement of the weak damping harmonic signals against the background noises. Ding & Chao (2015, GJI)'s Fig. 5 showed this point well, and we will not illustrate it here. Whether by multiplying the AR-$z$ spectra of multiple real time series, or by repeatedly adding random white noise to a single series to obtain the AR-$z$ product spectrum, the purpose is to make the noise variance for the empty bins to get reduced in principle through destructive interference, while the real signal's spectral peaks remain essentially unchanged.

[Figure]

**Figure R1**. The normalized Fourier, AR-z, and stabilized AR-$z$ spectra of a synthetic time series, which contains a 6-year signal and white noise.

**References**

Ding, H. and Chao, B. F.: Detecting harmonic signals in a noisy time series: The $z$-domain autoregressive (AR-z) spectrum, Geophys. J. Int., 201(3), 1287-1296, https://doi.org/10.1093/gji/ggv077, 2015.

Ding, H. and Chao, B. F.: The Slichter mode of the earth: Revisit with optimal stacking and autoregressive methods on full superconducting gravimeter data set, J. Geophys. Res. Solid Earth, 120, 7261-7272. https://doi.org/10.1002/2015JB012203, 2015.

Ding, H. and Chao, B. F.: Application of stabilized AR-$z$ spectrum in harmonic analysis for geophysics, J. Geophys. Res. Solid Earth, 123, 8249-8259, https://doi.org/10.1029/2018JB015890, 2018.